# COOPERATIVE SHEAF NEURAL NETWORKS

**André Ribeiro**
Getulio Vargas Foundation
andre.guimaraes@fgv.br

**Ana Luiza Tenório**
Getulio Vargas Foundation
ana.tenorio@fgv.br

**Juan Belieni**
Getulio Vargas Foundation
juanbelieni@gmail.com

**Amauri H. Souza**
Federal Institute of Ceará, $2\delta$ AI
amauriholanda@ifce.edu.br

**Diego Mesquita**
Getulio Vargas Foundation, $2\delta$ AI
diego.mesquita@fgv.br

## ABSTRACT

Sheaf neural networks (SNNs) leverage cellular sheaves to induce flexible diffusion processes on graphs, generalizing the diffusion mechanism of classical graph neural networks. While SNNs have been shown to cope well with heterophilic tasks and alleviate oversmoothing, we show that there is further room for improving sheaf diffusion. More specifically, we argue that SNNs do not allow nodes to independently choose how they cooperate with their neighbors, i.e., whether they convey and/or gather information to/from their neighbors. To address this issue, we first introduce the notion of cellular sheaves over directed graphs and characterize their in- and out-degree Laplacians. We then leverage our construction to propose Cooperative Sheaf Neural Network (CSNN). Additionally, we formally characterize its receptive field and prove that it allows nodes to selectively attend (listen) to arbitrarily far nodes while ignoring all others in their path, which is key to alleviating oversquashing. Our results on synthetic data empirically substantiate our claims, showing that CSNN can handle long-range interactions while avoiding oversquashing. We also show that CSNN performs strongly in heterophilic node classification and long-range graph classification benchmarks.

## 1 INTRODUCTION

Graph neural networks (GNNs) have become the standard models for an array of predictive tasks over networked data, with far-reaching applications in, e.g., physics simulation (Sanchez-Gonzalez et al., 2020), recommender systems (Ying et al., 2018), and molecular modeling (Duvenaud et al., 2015; Gilmer et al., 2017). Nonetheless, classical GNNs have well-known pitfalls. For instance, they typically struggle on heterophilic tasks (Zhu et al., 2020) — i.e., cases where connected nodes often belong to different classes or have dissimilar features. Furthermore, GNNs may also be susceptible to oversmoothing (Oono and Suzuki, 2020) and oversquashing (Alon and Yahav, 2021). Oversmoothing occurs when stacking multiple GNN layers yields increasingly similar node representations, whereas oversquashing refers to the loss of information when carrying information through increasingly long paths — due to the compression of exponentially growing information into fixed-size vectors.

A recent line of works (e.g., Hansen and Gebhart, 2020; Bodnar et al., 2022; Bamberger et al., 2025), which we henceforth refer to as Sheaf Neural Networks (SNNs), proposes modeling node interactions using cellular sheaves to achieve a principled solution to deal with oversmoothing and heterophilic tasks. A cellular sheaf $\mathcal{F}$ over an undirected graph associates (i) vector spaces $\mathcal{F}(i)$ and $\mathcal{F}(e)$, known as *stalks*, to each vertex $i$ and each edge $e$, and (ii) a linear map $\mathcal{F}_{i \triangleleft e}$, known as a *restriction map*, to each incident vertex-edge pair $i \triangleleft e$. These mathematical constructs induce a sheaf Laplacian which is governed by the restriction maps and generalizes the conventional Graph Laplacian.

In a parallel line of investigation, Finkelshtein et al. (2024) have recently shown that GNNs generally lack the flexibility to allow for nodes to individually select how they cooperate with their neighbors, i.e., choose whether they convey and/or gather information to/from their neighbors. This selective communication (also called cooperative behavior) is an especially desirable trait to tackle oversquash-

ing, as it allows controlling the amount of information flowing between nodes. A natural question ensues: *Can sheaf neural networks achieve cooperative behavior?*

In this paper, we provide a negative answer to this question. More precisely, for SNNs to zero out all the incoming information at a node $i$, they must set $\mathcal{F}_{i \trianglelefteq e} = 0$ for all incident edges $e$, which also implies the information flowing from $i$ is suppressed (see Figure 2). To circumvent this limitation, we introduce the notion of directed cellular sheaves (Definition 3.2) and define their in- and out-degree Laplacians (Definition 3.3). Leveraging these notions, we propose Cooperative Sheaf Neural Networks (CSNNs). Importantly, we show that cooperative behavior can be achieved using only a pair of restriction maps per node, which considerably increases computational efficiency compared to full sheaves — in which the amount of restriction maps increases linearly with the number of edges.

Our theoretical results show that CSNN allows for nodes to selectively listen to other arbitrarily distant nodes, which is a desirable trait to alleviate over-squashing. Our results on synthetic data specifically designed to induce over-squashing (Alon and Yahav, 2021) substantiate our claims, showcasing CSNNs' superior potential to handle long-range dependencies. In addition, extensive real-world experiments on 11 node-classification benchmarks and 2 long-range graph-classification tasks demonstrate that CSNN typically outperforms existing SNNs and cooperative GNNs.

In summary, our **contributions** are:

1. We introduce the notions of in- and out-degree Laplacians for cellular sheaves over directed graphs, which can be used to model asymmetric relationships between nodes. We treat undirected edges as a pair of directed ones and leverage these constructions to propose CSNN — provably extending the flexibility of sheaf diffusion to accommodate cooperative behavior;

2. We provide a theoretical analysis of CSNN, showing that: ($a$) for each layer $t$ in CSNN, nodes may be affected by information from nodes at distance up to $2t$-hop neighbors (Proposition 4.2), instead of up to $t$-hop neighbors in usual GNNs; and ($b$) there exist restriction maps which make the embedding of a node $i$ at layer $t$ highly sensitive to the initial feature of a node $j$, where $t$ is also the distance between $i$ and $j$ (Proposition 4.3).

3. We carry an extensive experimental campaign to validate the effectiveness of CSNNs, encompassing both synthetic and real-world tasks. Our experiments on synthetic data show that CSNNs is remarkably capable of mitigating over-squashing and modeling long-range dependencies. Meanwhile, results on over 13 real-world tasks show CSNNs typically outperform prior sheaf-based models and cooperative GNNs.

## 2 BACKGROUND

For the sake of completeness, here we provide a summary of core concepts concerning cellular sheaves over undirected graphs. We also briefly discuss neural sheaf diffusion and cooperative GNNs.

In this work, we denote an undirected graph by a tuple $G = (V, E)$ where $V$ is a set of vertices (or nodes) and $E$ is a set of unordered pairs of (distinct) vertices, called edges, with $n = |V|$ and $m = |E|$. We denote the neighbors of a node $i$ in $G$ by $N(i) = \{j : \{i, j\} \in E\}$. Next, Definition 2.1 introduces the notion of cellular sheaves over undirected graphs.

**Definition 2.1.** A **cellular sheaf** $(G, \mathcal{F})$ over a (undirected) graph $G = (V, E)$ associates:

1. Vector spaces $\mathcal{F}(i)$ to each vertex $i \in V$ and $\mathcal{F}(e)$ to each edge $e \in E$, called **stalks**.
2. Linear maps $\mathcal{F}_{i \trianglelefteq e} : \mathcal{F}(i) \to \mathcal{F}(e)$ to each incident vertex-edge pair $i \trianglelefteq e$, called **restriction maps**.

Hereafter, we assume all vertex and edge stalks are isomorphic to $\mathbb{R}^d$. If all restriction maps are equal to the identity map, we say the cellular sheaf is constant. Moreover, if $d = 1$, the sheaf is said trivial.

Given the importance of the Laplacian operator for graph representation learning, it is instrumental to define the Laplacian for undirected cellular sheaves – a key concept in the design of SNNs. Towards this end, we introduce in Definition 2.2 the spaces of 0- and 1-cochains.

**Definition 2.2.** The **space of 0-cochains**, denoted by $C^0(G, \mathcal{F})$, and the **space of 1-cochains**, $C^1(G, \mathcal{F})$, of a cellular sheaf $(G, \mathcal{F})$ are given by

$$C^0(G, \mathcal{F}) = \bigoplus_{i \in V} \mathcal{F}(i) \text{ and } C^1(G, \mathcal{F}) = \bigoplus_{e \in E} \mathcal{F}(e). \tag{1}$$

where $\oplus$ denotes the (external) direct sum.

Now, for each $e \in E$ choose an orientation $e = i \to j$ and consider the coboundary operator $\delta : C^0(G, \mathcal{F}) \to C^1(G, \mathcal{F})$ defined by $(\delta \mathbf{X})_e = \mathcal{F}_{j \trianglelefteq e} \mathbf{x}_j - \mathcal{F}_{i \trianglelefteq e} \mathbf{x}_i$. Then, the sheaf Laplacian is defined by $L_\mathcal{F} = \delta^\top \delta$. If $\mathcal{F}$ is the trivial sheaf, $\delta^\top$ can be seen as the incidence matrix, recovering the $n \times n$ graph Laplacian. A more explicit way to describe the Laplacian is the following:

**Definition 2.3.** The **sheaf Laplacian** of a cellular sheaf $(G, \mathcal{F})$ is the linear operator $L_\mathcal{F} : C^0(G, \mathcal{F}) \to C^0(G, \mathcal{F})$ that, for a 0-cochain $\mathbf{X} \in C^0(G, \mathcal{F})$, outputs

$$L_\mathcal{F}(\mathbf{X})_i := \sum_{i,j \trianglelefteq e} \mathcal{F}_{i \trianglelefteq e}^\top \left( \mathcal{F}_{i \trianglelefteq e} \mathbf{x}_i - \mathcal{F}_{j \trianglelefteq e} \mathbf{x}_j \right) \qquad \forall i \in V. \tag{2}$$

The Laplacian $L_\mathcal{F}$ can also be seen as a positive semidefinite matrix with diagonal blocks $L_{ii} = \sum_{i \trianglelefteq e} \mathcal{F}_{i \trianglelefteq e}^\top \mathcal{F}_{i \trianglelefteq e}$ and non-diagonal blocks $L_{ij} = L_{ij}^\top = -\mathcal{F}_{i \trianglelefteq e}^\top \mathcal{F}_{j \trianglelefteq e}$.

To build intuition around Definition 2.1, we may interpret the node stalks $\mathcal{F}(i)$ as the space of private opinions held by an individual $i$, following the perspective of Hansen and Ghrist (2021). For an edge $e$ connecting nodes $i$ and $j$, the stalk $\mathcal{F}(e)$ corresponds to the public opinions exchanged between them.

That being said, note that $\ker L_\mathcal{F} = \{ \mathbf{X} \in C^0(G, \mathcal{F}) \mid \mathcal{F}_{i \trianglelefteq e} \mathbf{x}_i = \mathcal{F}_{j \trianglelefteq e} \mathbf{x}_j \, \forall e = \{i, j\} \in E(G) \}$. This can be understood as the space of public agreement between all pairs of neighboring nodes $i$ and $j$. Note that $i$ and $j$ can have distinct opinions about the same topic on their respective private opinion spaces $\mathcal{F}(i)$ and $\mathcal{F}(j)$; however, when they publicly discuss this topic, they may prefer to not manifest their true opinion. Alternatively, since the edge stalks may be different from the node stalks, some topics of the private opinion spaces do not need to be discussed at all. In both cases, the apparent consensus lies in $\operatorname{Ker} L_\mathcal{F}$.

**Vector bundles.** When restriction maps are orthogonal, we call the sheaf a vector bundle. In this case, $L_\mathcal{F}(\mathbf{X})_i := \sum_{i,j \trianglelefteq e} \left( \mathbf{x}_i - \mathcal{F}_{i \trianglelefteq e}^\top \mathcal{F}_{j \trianglelefteq e} \mathbf{x}_j \right)$, for any $i \in V$. Flat vector bundles are special cases of vector bundles in which we assign an orthogonal map $\mathbf{O}_i$ to each node $i$ and set $\mathcal{F}_{i \trianglelefteq e} = \mathbf{O}_i$ for all $e$ incident to $i$. This entails $L_\mathcal{F}(\mathbf{X})_i := \sum_{i,j \trianglelefteq e} \left( \mathbf{x}_i - \mathbf{O}_i^\top \mathbf{O}_j \mathbf{x}_j \right)$, for any $i \in V$. Note that flat vector bundles only comprise $n$ restriction maps as opposed to $2m$ maps in general cellular sheaves. Prior works (Bodnar et al., 2022; Bamberger et al., 2025) have leveraged these simpler constructions to propose computationally efficient sheaf-based neural networks.

**Neural Sheaf Diffusion (NSD).** Bodnar et al. (2022) introduce NSD building on Euler iterations of the heat equation induced by $\Delta_\mathcal{F}$, i.e., $\dot{\mathbf{X}} = -\Delta_\mathcal{F} \mathbf{X}$. First, we project the initial node features $\mathbf{X}$ into $h$ channels using an MLP $\eta$, i.e., $\mathbf{X}_0 = \eta(\mathbf{X}) \in \mathbb{R}^{nd \times h}$. Then, NSD recursively computes

$$\mathbf{X}_{t+1} = (1 + (\mathbf{1}_{n \times h} \otimes \varepsilon)) \odot \mathbf{X}_t - \sigma(\Delta_{\mathcal{F}(t)}(\mathbf{I} \otimes \mathbf{W}_{1,t}) \mathbf{X}_t \mathbf{W}_{2,t}), \tag{3}$$

where $\mathbf{1}_{n \times h}$ is an $n \times h$ matrix of ones, $\Delta_{\mathcal{F}(t)}$ is the sheaf Laplacian at layer $t$, $\varepsilon \in [-1, 1]^d$ is a (learned) vector scaling the features along each stalk dimension, $\sigma$ is an element-wise non-linearity, and $\mathbf{W}_{1,t} \in \mathbb{R}^{d \times d}$, $\mathbf{W}_{2,t} \in \mathbb{R}^{h \times h}$ are weight matrices. Importantly, the restriction maps which govern $\Delta_{\mathcal{F}(t)}$ are learned in an end-to-end fashion, alongside $\mathbf{W}_{1,t}$ and $\mathbf{W}_{2,t}$.

**Cooperative GNNs.** Finkelshtein et al. (2024) recently proposed flexibilizing message-passing GNNs by treating nodes as players that can choose how they cooperate with their neighbors. More specifically, cooperative GNNs employ an auxiliary GNN, called the *action* network, that decides individually how each node partakes in the message passing of the base GNN, or *environment* network. The action network decides whether each node only propagates information (PROPAGATE), only gathers information from neighbors (LISTEN), does none (ISOLATE) or does both (STANDARD). Cooperative GNNs learn the action and environment GNNs simultaneously, using the straight-through Gumbel-Softmax estimator to propagate gradients through the discrete actions of the action network.

## 3 CELLULAR SHEAVES FOR DIRECTED GRAPHS

We kick off this section addressing our initial research question: Can SNNs achieve cooperative behavior? Recall that communication between nodes in an SNN is governed by its sheaf Laplacian, which is induced by the restriction maps. Thus, in SNNs, picking a state among PROPAGATE, LISTEN, ISOLATE, and STANDARD node $i$ translates to choosing a suitable configuration for its respective restriction maps. The following result says that SNNs cannot fully alternate between these action.

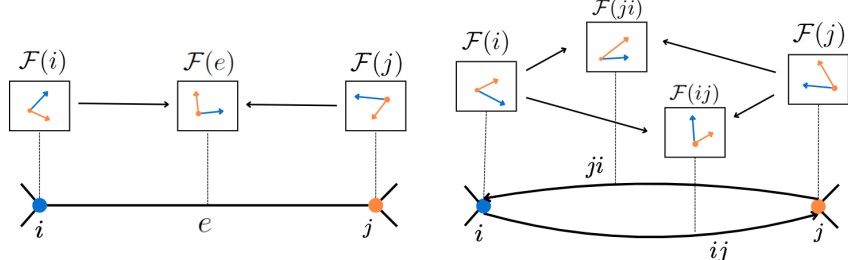

Figure 1: On the left, a cellular sheaf shown for a single edge of an undirected graph with stalks isomorphic to $\mathbb{R}^2$. The restriction maps $\mathcal{F}_{i \trianglelefteq e}, \mathcal{F}_{j \trianglelefteq e}$ move the vector features between these spaces. On the right, the analogous situation for a sheaf on a single pair of directed edges. Then there are four, possibly distinct, restriction maps $\mathcal{F}_{i \trianglelefteq ij}, \mathcal{F}_{i \trianglelefteq ji}, \mathcal{F}_{j \trianglelefteq ij}, \mathcal{F}_{j \trianglelefteq ji}$.

**Proposition 3.1.** *Let $i \in V$. If $L_{\mathcal{F}}(\mathbf{X})_i$ does not depend on $\mathbf{x}_j$ for any $j \in V$ neighbor of $i$, then $L_{\mathcal{F}}(\mathbf{X})_j = 0$ or $L_{\mathcal{F}}(\mathbf{X})_j = \sum_{j, i \trianglelefteq e} \mathcal{F}_{j \trianglelefteq e}^\top \mathcal{F}_{j \trianglelefteq e} \mathbf{x}_j$.*

Put plainly, Proposition 3.1 states that the sheaf diffusion provides a framework where a node $i$ that does not LISTEN (since it does depend on $j$) must not PROPAGATE (since the update of $j$ does not depend on $i$), independently of the action that $j$ takes. In other words, PROPAGATE implies LISTEN, which means it collapses to ISOLATE (see Figure 2).

We can circumvent this limitation by treating undirected edges as a pair of directed ones, creating an additional channel of communication between nodes. To accommodate directed edges (i.e., $E \subseteq V \times V$), we propose using cellular sheaves over directed graphs.

Cellular sheaves over directed graphs must distinguish the restriction map where $i$ is the source node of an edge from the restriction map where $i$ is the target node of an edge. Therefore, we change the edge notation from $e$ to $ij$ and $ji$ to make this distinction explicit. See Figure 1 for an illustration.

**Definition 3.2.** A **cellular sheaf** $(G, \mathcal{F})$ over a (directed) graph $G = (V, E)$ associates:
1. Vector spaces $\mathcal{F}(i)$ to each vertex $i \in V$ and $\mathcal{F}(ij)$ to each edge $ij \in E$, called **stalks**.
2. A linear map $\mathcal{F}_{i \trianglelefteq ij} : \mathcal{F}(i) \to \mathcal{F}(ij)$ for each incident vertex-edge pair $i \trianglelefteq ij$ and a linear map $\mathcal{F}_{i \trianglelefteq ji} : \mathcal{F}(i) \to \mathcal{F}(ji)$ for each incident vertex-edge pair $i \trianglelefteq ji$, called **restriction maps**.

For simplicity, again, we henceforth assume all node and edge stalks are $d$-dimensional.

We are now left with the task of defining sheaf Laplacians that can be used for information diffusion in directed graphs. For directed graphs, it is common to define both in- and out-degree Laplacians (Agaev and Chebotarev, 2005). Given a directed graph with possibly asymmetric adjacency matrix $A$, the out-degree Laplacian is $L^{\text{out}} := D^{\text{out}} - A$ and the in-degree Laplacian is $L^{\text{in}} := D^{\text{in}} - A$, with $D^{\text{in}}$ and $D^{\text{out}}$ denoting the diagonal matrices containing in- and out-degree of nodes in $V(G)$. Definition 3.3 below generalizes these notions to sheaves over directed graphs.

**Definition 3.3.** The **out-degree sheaf Laplacian** of a cellular sheaf $(G, \mathcal{F})$ is the linear operator $L_{\mathcal{F}}^{\text{out}} : C^0(G, \mathcal{F}) \to C^0(G, \mathcal{F})$ that, for a 0-cochain $\mathbf{X} \in C^0(G, \mathcal{F})$, outputs

$$L_{\mathcal{F}}^{\text{out}}(\mathbf{X})_i := \sum_{j \in N(i)} \left( \mathcal{F}_{i \trianglelefteq ij}^\top \mathcal{F}_{i \trianglelefteq ij} \mathbf{x}_i - \mathcal{F}_{i \trianglelefteq ji}^\top \mathcal{F}_{j \trianglelefteq ji} \mathbf{x}_j \right), \qquad \forall i \in V. \tag{4}$$

The **in-degree sheaf Laplacian** of a cellular sheaf $(G, \mathcal{F})$ is the linear operator $L_{\mathcal{F}}^{\text{in}} : C^0(G, \mathcal{F}) \to C^0(G, \mathcal{F})$ that, for a 0-cochain $\mathbf{X} \in C^0(G, \mathcal{F})$, outputs

$$L_{\mathcal{F}}^{\text{in}}(\mathbf{X})_i := \sum_{j \in N(i)} \left( \mathcal{F}_{i \trianglelefteq ji}^\top \mathcal{F}_{i \trianglelefteq ji} \mathbf{x}_i - \mathcal{F}_{i \trianglelefteq ij}^\top \mathcal{F}_{j \trianglelefteq ij} \mathbf{x}_j \right), \qquad \forall i \in V. \tag{5}$$

We note that if $(G, \mathcal{F})$ is the trivial sheaf, then $L_{\mathcal{F}}^{\text{out}} = (L^{\text{out}})^\top$ and $L_{\mathcal{F}}^{\text{in}} = L^{\text{in}}$.

**Flat vector bundles over directed graphs.** We can also improve the parameter efficiency of cellular sheaves over directed graphs using flat vector bundles. Since the graphs are directed, we need to distinguish between edges with identical endpoints but with different orientations. Thus, for each node $i$, we assign a source conformal map $\mathbf{S}_i$ and a target conformal map $\mathbf{T}_i$, and set $\mathcal{F}_{i \trianglelefteq ij} = \mathbf{S}_i$ and $\mathcal{F}_{i \trianglelefteq ji} = \mathbf{T}_i$ for all neighbors $j$ of $i$.

## 4 COOPERATIVE SHEAF NEURAL NETWORKS

In this section, we leverage the sheaf Laplacians in Definition 3.3 to propose Cooperative Sheaf Neural Networks (CSNNs), an SNN which allows nodes to independently to decide how they participate in message diffusion, choosing whether to broadcast their information and/or to listen from the neighbors. To exploit the asymmetric communication induced by sheaves over directed graphs, we convert our input undirected graph into a directed one by replacing undirected edges with a pair of directed ones.

We design CSNN's diffusion mechanism by composing the out-degree and the transposed in-degree sheaf Laplacians. In practice, we use their normalized versions

$$\Delta_{\mathcal{F}}^{\text{out}} = D_{\text{out}}^{-\frac{1}{2}} L_{\mathcal{F}}^{\text{out}} D_{\text{out}}^{-\frac{1}{2}} \quad \text{and} \quad (\Delta_{\mathcal{F}}^{\text{in}})^{\top} = D_{\text{in}}^{-\frac{1}{2}} (L_{\mathcal{F}}^{\text{in}})^{\top} D_{\text{in}}^{-\frac{1}{2}},$$

where $D_{\text{in}}, D_{\text{out}}$ are the block-diagonals of the in and out Laplacians, respectively.

> We define a CSNN layer by augmenting the Euler discretization of our novel heat equation $\dot{\mathbf{X}} = (\Delta_{\mathcal{F}}^{in})^{\top} \Delta_{\mathcal{F}}^{out} \mathbf{X}$ with linear transformations and a nonlinear activation function $\sigma$:
>
> $$\mathbf{X}_{t+1} = (1 + (\mathbf{1}_{n \times h} \otimes \varepsilon)) \odot \mathbf{X}_t - \sigma((\Delta_{\mathcal{F}(t)}^{in})^{\top} \Delta_{\mathcal{F}(t)}^{out} (\mathbf{I}_n \otimes \mathbf{W}_{1,t}) \mathbf{X}_t \mathbf{W}_{2,t}), \quad (6)$$
>
> where $\mathbf{1}_{n \times h}$ is an $n$-by-$h$ matrix of ones, $\varepsilon \in [-1,1]^{d \times 1}$, and $\mathbf{W}_{1,t} \in \mathbb{R}^{d \times d}$ and $\mathbf{W}_{2,t} \in \mathbb{R}^{h \times h}$ are learned matrices responsible for mixing node features and channels, respectively.

**Efficient implementation.** For computational efficiency, we use flat vector bundles to define both the in- and out- degree sheaf Laplacians. More precisely, for each node $i$, we define a source conformal map $\mathbf{S}_i$ and a target conformal map $\mathbf{T}_i$ for all neighbor $j$ of $i$. Thus, out-degree sheaf Laplacian simplifies to

$$L_{\mathcal{F}}^{\text{out}}(\mathbf{X})_i := \sum_{j \in N(i)} \left( \mathbf{S}_i^{\top} \mathbf{S}_i \mathbf{x}_i - \mathbf{T}_i^{\top} \mathbf{S}_j \mathbf{x}_j \right), \quad (7)$$

while the transpose of the in-degree sheaf Laplacian is

$$((L_{\mathcal{F}}^{\text{in}})^{\top}(\mathbf{X}))_i := \sum_{j \in N(i)} \left( \mathbf{T}_i^{\top} \mathbf{T}_i \mathbf{x}_i - \mathbf{T}_i^{\top} \mathbf{S}_j \mathbf{x}_j \right). \quad (8)$$

Note these matrices have a block structure, with diagonals $(L_{\mathcal{F}}^{\text{in}})_{ii}^{\top} = \sum \mathbf{T}_i^{\top} \mathbf{T}_i$, $(L_{\mathcal{F}}^{\text{out}})_{ii} = \sum \mathbf{S}_i^{\top} \mathbf{S}_i$, and remaining blocks $(L_{\mathcal{F}}^{\text{in}})_{ij}^{\top} = (L_{\mathcal{F}}^{\text{out}})_{ij} = -\mathbf{T}_i^{\top} \mathbf{S}_j$.

We point out that, since conformal maps are of the form $\mathbf{S}_i = C_{\mathbf{S}_i} Q_i$ and $\mathbf{T}_i = C_{\mathbf{T}_i} R_i$, for some orthogonal matrices $Q_i$, $R_i$ and scalars $C_{\mathbf{S}_i}$, $C_{\mathbf{T}_i}$, computing their inverses and normalizing the Laplacian becomes trivial. In this case, block scaling simplifies to a block matrix of scalars time identity and the normalization is both numerically stable and computationally efficient.

The first step of each layer $t$ consists of computing the conformal maps $\mathbf{T}_{i,t}$ and $\mathbf{S}_{i,t}$. We do this through learnable functions $\mathbf{S}_{i,t} = \eta(G, \mathbf{X}_t, i)$ and $\mathbf{T}_{i,t} = \phi(G, \mathbf{X}_t, i)$, $\forall i \in V$. As in prior works on SNNs Bodnar et al. (2022); Bamberger et al.

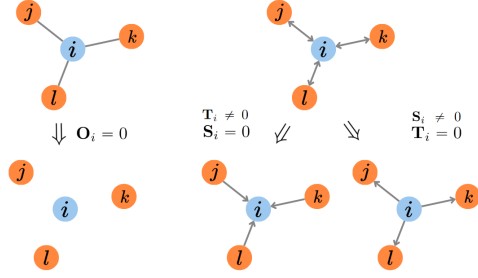

Figure 2: $\mathbf{O}_i = 0$ creates the effect of isolating the node $i$. Directed edges allow for LISTEN and PROPAGATE separately.

(2025), we use neural networks to learn the restriction maps. In addition, we use Householder reflections (Mhammedi et al., 2017; Obukhov, 2021) to compute orthogonal maps and multiply them by a learned positive constant for each node.

### 4.1 ANALYSIS

We now show that CSNN can achieve cooperative behavior, characterize its receptive field, and prove that appropriate conformal maps can help CSNNs handle long-range interactions.

We note that CSNN allows each node $i$ drift from the STANDARD behavior ($\mathbf{S}_i, \mathbf{T}_i \neq 0$) by zeroing-out its conformal maps. Setting $\mathbf{T}_i \neq 0$ drives $i$ to LISTEN. Setting $\mathbf{S}_i \neq 0$ corresponds to PROPAGATE. Finally, $\mathbf{S}_i = \mathbf{T}_i = 0$ implies ISOLATE.

We highlight the importance of considering the directions: for undirected graphs, there is a single map $\mathbf{O}_i$ for each node $i$, where $\mathbf{O}_i = 0$ could only mean that $i$ does not communicate at all. In other words, the possible actions are only STANDARD and ISOLATE. We illustrate this in Figure 2.

Thus, to show a model achieves cooperative behavior, we must ensure the following: (a) if $i$ does not listen, then its update cannot depend on $\mathbf{x}_j$, $\forall j \in V, j \neq i$; and (b) if $i$ has a non-propagating neighbor $k$, then its update cannot depend on $\mathbf{x}_k$. Proposition 4.1 shows that CSNN satisfies these conditions, while Figure 3 illustrates them and delineates limitations of the NSD model.

**Proposition 4.1.** *If the target map $\mathbf{T}_i$ is zero, then $((L_{\mathcal{F}}^{in})^{\top} L_{\mathcal{F}}^{out}(\mathbf{X}))_i = 0$. If the source map $\mathbf{S}_k = 0$ for some neighbor $k$ of $i$, then $((L_{\mathcal{F}}^{in})^{\top} L_{\mathcal{F}}^{out}(\mathbf{X}))_i$ does not depend on $\mathbf{x}_k$.*

Moreover, our model has the ability to reach longer distances. In most GNNs, if $t$ is the distance between two nodes $i$ and $j$, then they can only communicate after $t$ layers. CSNN enables communication between these nodes after $\lceil t/2 \rceil$ layers.

**Proposition 4.2.** *In each layer $t$, the features of a node can be affected by the features of nodes up to $2t$-hops.*

We also show in Proposition 4.3 that CSNNs with $t$ layers are capable of making $i$ and $j$ communicate while ignoring all the other nodes on a path from $i$ to $j$ such that $|j - i| \leq t$. This feature is an asset to handle over-squashing in long range tasks, allowing CSNN to selectively tend to information from distant nodes. Example 4.4 illustrates this result in a four-node graph.

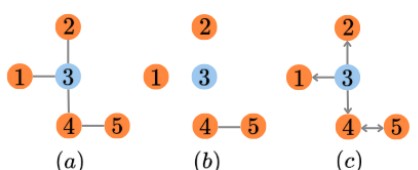

(a)     (b)     (c)

Figure 3: Given a graph $(a)$ we illustrate the consequences of preventing node 3 from listening. For NSD $(b)$, this means $L_{\mathcal{F}}(\mathbf{X})_3$ must not depend on $\mathbf{x_j}$, for $j = 1, 2, 4$ implying $\mathcal{F}_{j \trianglelefteq e} = 0$ and leading to $L_{\mathcal{F}}(\mathbf{X})_j$ not depending on $\mathbf{x_3}$, preventing node 3 from propagating information. In CSNN $(c)$, we can set $\mathbf{T}_3 = 0$. Provided $\mathbf{S}_3 \neq 0$, outbound communication is possible.

**Proposition 4.3.** *Let $i$ and $j$ be nodes at a distance $t$. In CSNN, $i$ can learn to ignore all the $t - 1$ nodes in the shortest path from $i$ to $j$ while receiving the information from $j$ in the $t$-layer. Moreover, if we choose a path with $n > t - 1$ nodes between $i$ and $j$, then $i$ receives the information from $j$ in the $(n + 1)$-layer.*

**Example 4.4.** *Consider a directed graph with vertex set $V = \{1, 2, 3, 4\}$ and edge set $E = \{(1, 2), (2, 3), (3, 4), (4, 3), (3, 2), (2, 1)\}$. We follow the proof of Proposition 4.3 to show we can propagate a message from node 4 to node 1, while the latter ignores all remaining nodes. We denote the target/source map of node $i$ at layer $t$ by $\mathbf{T}_{i,t}$ and $\mathbf{S}_{i,t}$ respectively. To achieve our desired result:*

a. *In the first layer, must have zero source and target maps except for $\mathbf{T}_{3,1}$ and $\mathbf{S}_{4,1}$. A simple verification gives that $((L_{\mathcal{F}}^{in})^{\top} L_{\mathcal{F}}^{out}(\mathbf{X}))_k = 0$, for all $k \neq 3$. So $((L_{\mathcal{F}}^{in})^{\top} L_{\mathcal{F}}^{out}(\mathbf{X}))_k = 0$, for all $k \neq 3$. If $k = 3$, then $((L_{\mathcal{F}}^{in})^{\top} L_{\mathcal{F}}^{out}(\mathbf{X}))_3 = -2\mathbf{T}_{4,1}^{\top} \mathbf{S}_{4,1} \mathbf{x}_4^{(0)}$, with $\mathbf{x}_k^{(t)}$ denoting the feature vector of $k$ at layer $t$, thus $\mathbf{x}_4^{(0)}$ will be the only feature vector to influence $\mathbf{x}_3^{(1)}$;*

b. *In the second layer, we must have that all source and target maps are zero except for $\mathbf{T}_{2,2}$ and $\mathbf{S}_{3,2}$. Then $((L_{\mathcal{F}}^{in})^{\top} L_{\mathcal{F}}^{out}(\mathbf{X}))_k = 0$, for all $k \neq 2$ and $((L_{\mathcal{F}}^{in})^{\top} L_{\mathcal{F}}^{out}(\mathbf{X}))_2 = -2\mathbf{T}_{2,2}^{\top} \mathbf{S}_{3,2} \mathbf{x}_3^{(1)}$. Thus $\mathbf{x}_3^{(1)}$ will be the only feature vector to influence $\mathbf{x}_2^{(2)}$;*

c. *In the third layer, all source and target maps must be zero except for $\mathbf{T}_{1,3}$ and $\mathbf{S}_{2,3}$. Then $((L_{\mathcal{F}}^{in})^{\top} L_{\mathcal{F}}^{out}(\mathbf{X}))_k = 0$, for all $k \neq 1$ and $((L_{\mathcal{F}}^{in})^{\top} L_{\mathcal{F}}^{out}(\mathbf{X}))_1 = -2\mathbf{T}_{1,3}^{\top} \mathbf{S}_{2,3} \mathbf{x}_2^{(3)}$. Thus $\mathbf{x}_1^{(3)}$ is influenced only by $\mathbf{x}_2^{(2)}$, which was influenced only by $\mathbf{x}_3^{(1)}$, which is influenced only by $\mathbf{x}_4^{(0)}$.*

*Consequently, $\mathbf{x}_1^{(3)}$ is affected by $\mathbf{x}_4^{(0)}$ while ignoring the features of all other nodes in all other layers.*

*This configuration shows that although CSNN can achieve 2-hop neighbors, it can also refrain from this behavior to only access 1-hop neighbors per layer. Moreover, this flexibility indicates there are multiple forms to stablish communication between two distant nodes while ignoring others.*

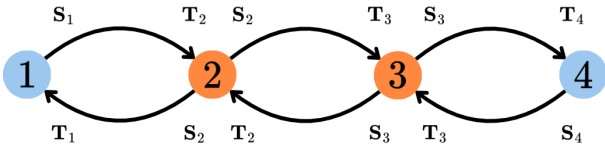

Figure 4: Illustration of Example 4.4. At layer $t$, we consider that all maps but $\mathbf{T}_{4-t,t}$ and $\mathbf{S}_{4-(t-1),t}$ are 0, enabling the flow of information from right to left following the bottom edges.

Observe that the derivative of $\mathbf{x}_1^{(3)}$ in relation to $\mathbf{x}_4^{(0)}$ can be as high as the values of the non-zero $\mathbf{T}_i$ and $\mathbf{S}_i$ permit. This shows our model can mitigate over-squashing, which refers to the failure of an information propagating to distance nodes. Di Giovanni et al. (2023) and Topping et al. (2022), studied over-squashing in message passing neural networks through a bound on the Jacobian $\left|\partial\mathbf{x}_i^{(t)}/\partial\mathbf{x}_j^{(0)}\right| \leqslant c^t \hat{A}_{ij}^t$, where $t$ is the layer, $c$ is a constant that depends on the architecture of the model, and $\hat{A}$ the normalized adjacency matrix. Moreover, over-squashing occurs when we have a small derivative $\partial\mathbf{x}_i^{(t)}/\partial\mathbf{x}_j^{(0)}$, since it means that after $t$ layers, the feature at $i$ is mostly insensitive to the information initially contained at $j$, i.e., the information was not propagated properly. Proposition 4.3 states that the feature at node $i$ can be sensitive to the information initially contained at node $j$, independently of the distance, given enough layers and an appropriate configuration of restriction maps. Consequently, Proposition 4.3 suggests that the restriction maps regulating the sensibility between distant nodes can provide higher upper bounds to $\partial\mathbf{x}_i^{(t)}/\partial\mathbf{x}_j^{(0)}$ while decreasing the value of $\partial\mathbf{x}_i^{(t)}/\partial\mathbf{x}_k^{(0)}$ for other nodes $k$.

## 5 RELATED WORKS

**Cooperative GNNs.** Finkelshtein et al. (2024) were the first to propose flexibilizing message passing by allowing nodes to choose how they cooperate with each other. Each layer of their model, CO-GNN, employs an additional GNN that chooses an action for each node. While CO-GNNs can be employed with arbitrary base and action networks, their main caveat is that training can become increasingly difficult as these networks become more complex — the grid of hyper-parameters grows considerably and the stochastic nature of the action network may affect model selection. Different from CO-GNN, our CSNN does not rely on discrete actions and can smoothly modulate between cooperative behavior patterns.

**Sheaf Neural Networks.** Besides works on SNNs for graph data with real-valued node features, recent works have expanded the literature to accommodate heterogeneous edge types (Braithwaite et al., 2024), hypergraphs (Duta et al., 2023), nonlinear Sheaf Laplacians (Zaghen et al., 2024), and node features living on Riemann manifolds (Battiloro et al., 2024). While recent works on SNNs learn restriction maps in an end-to-end fashion, there are also prior works in which they are manually constructed (Hansen and Ghrist, 2019) or computed as a pre-processing step (Barbero et al., 2022).

**Quiver Laplacians.** Sumray et al. (2024) propose sheaf Laplacians over quivers (directed graphs w/ self-loops) to improve feature selection on tabular data, with no learning component involved. The in- and out-Laplacians we defined here are not particular cases of the Laplacians over quivers. The former are positive semi-definite matrices, while our Laplacians may have complex eigenvalues with negative real parts.

## 6 EXPERIMENTS

We provide both synthetic and real-world experiments to evaluate the performance of CSNN, including node- and graph-level prediction tasks. Section 6.1 assess CSNN's capacity to circumvent over-squashing using the NeighborsMatch benchmark proposed by Alon and Yahav (2021). Section 6.2 presents experiments on eleven node classification tasks, showcasing the effectiveness of CSNN for heterophilic graphs. Finally, Section 6.3 consider the Peptides datasets from the Long Range Graph Benchmark (Dwivedi et al., 2022) to substantiate the capability of our model to mitigate under-reaching and over-squashing on real-world graph

classification. We also provide additional experiments in Appendix B. Our code can be found at `https://github.com/andrerg02/Cooperative-Sheaf-NN`.

## 6.1 OVER-SQUASHING

In order to verify our theoretical results on the capacity of CSNN to alleviate over-squashing, we reproduce the NeighborsMatch problem proposed by Alon and Yahav (2021), using the same framework. The datasets consist of binary trees of depth $r$, with the root node as the target, the leaves containing its possible labels, and the leaf with the same number of neighbors as the target node containing its true label. We provide the parameters used for this task in Appendix C.

Figure 5 shows GCN (Kipf and Welling, 2017) and GIN (Xu et al., 2019) fail to fit the datasets starting from $r = 4$ and GAT (Veličković et al., 2018) and GGNN (Li et al., 2016) fail to fit the datasets starting from $r = 5$. These models suffer from over-squashing and are not able to distinguish between different training examples, while the CSNN model reaches perfect accuracy for all tested $r$.

Alon and Yahav (2021) argue that the difference in performance for the GNNs are related to how node features are updated: on one hand, GCN and GIN aggregate all neighbor information before combining it with the representation of the target node, forcing them to compress all incoming information into a single vector. On the other hand, GAT uses attention to selectively weigh messages based on the representation of the target node, allowing it (to some degree) to filter out irrelevant edges and only compress information from a subset of the neighbors. So models like GAT (and GGNN) that compress less information per step can handle higher $r$ better than GCN and GIN.

This experiment shows that CSNN is more efficient in ignoring irrelevant nodes and can avoid loosing relevant information. Moreover, Proposition 4.3 provides theoretical support for this result, as it states that there are choices of parameters for which the model can listen *only* to the nodes along a path between distant nodes $i$ and $j$, enabling selective communication to diminish noise impact.

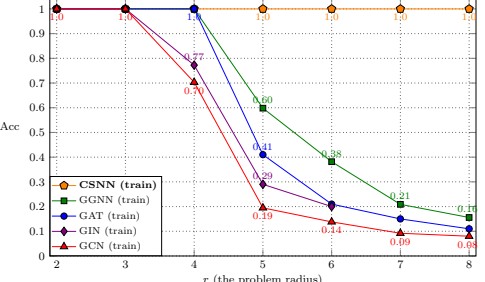

Figure 5: Accuracy for increasing tree depths in the NeighborsMatch task. CSNN consistently achieves 100% accuracy for all values of $r$.

**Comparison against other sheaf models.** Notably, CSNN outperforms other sheaf methods in this task. BuNN reports 100% accuracy until $r = 6$. Then it drops to 71% and 42% for $r = 7$ and $r = 8$, respectively, as reported in Bamberger et al. (2025). For NSD with orthogonal maps, we obtained 100% accuracy when $r = 2$, 91% for $r = 3$, and then a sharp drop to 5% when $r = 4$.

## 6.2 NODE CLASSIFICATION

**Datasets.** We evaluate our model on the five recently proposed heterophilic graphs from Platonov et al. (2023), and also on six classical ones for which benchmarking results can be found in Pei et al. (2020); Rozemberczki et al. (2021); Tang et al. (2009). As pointed out in Platonov et al. (2023), the datasets Squirrel and Chameleon have many duplicate nodes, which may lead to data leakage. Following their guidelines, we use their cleaned version of these datasets to ensure a meaningful evaluation. For binary classification datasets, we report AUROC, while for multiclass datasets we report accuracy.

**Experimental setting.** As baselines for benchmarks in Platonov et al. (2023), we use GCN (Kipf and Welling, 2017), GraphSAGE (Hamilton et al., 2017), GAT (Veličković et al., 2018) and GT (Shi et al., 2021), together with the variations GAT-sep and GT-sep, which concatenate the representation of a node to the mean of its neighbors instead of summing them (Zhu et al., 2020). These are all classical baselines used in Platonov et al. (2023) to compare against GNN architectures specifically developed for heterophilic settings, and that achieve the best performance in most cases. We also compare CSNN against recent models such as CO-GNN, NSD, and BuNN (Bamberger et al., 2025). Results for BuNN on Squirrel, Chameleon, and the datasets in Table 2 are not available, since we do not have access to their code.

Table 1: Performance comparison on datasets from Platonov et al. (2023). AUROC is reported for minesweeper, tolokers and questions, accuracy is reported for the remaining datasets. CSNN is the best-performing method in 6 out of 7 datasets.

| Model | roman-empire | amazon-ratings | minesweeper | tolokers | questions | squirrel | chameleon |
| Edge Homophily | 0.05 | 0.38 | 0.68 | 0.59 | 0.84 | 0.20 | 0.23 |
|---|---|---|---|---|---|---|---|
| GCN | 73.69 ± 0.74 | 48.70 ± 0.63 | 89.75 ± 0.52 | 83.64 ± 0.67 | 76.09 ± 1.27 | 39.47 ± 1.47 | 40.89 ± 4.12 |
| SAGE | 85.74 ± 0.67 | 53.63 ± 0.39 | 93.51 ± 0.57 | 82.43 ± 0.44 | 76.44 ± 0.62 | 36.09 ± 1.99 | 37.77 ± 4.14 |
| GAT | 80.87 ± 0.30 | 49.09 ± 0.63 | 92.01 ± 0.68 | 83.70 ± 0.47 | 77.43 ± 1.20 | 35.62 ± 2.06 | 39.21 ± 3.08 |
| GAT-sep | 88.75 ± 0.41 | 52.70 ± 0.62 | 93.91 ± 0.35 | 83.78 ± 0.43 | 76.79 ± 0.71 | 35.46 ± 3.10 | 39.26 ± 2.50 |
| GT | 86.51 ± 0.73 | 51.17 ± 0.66 | 91.85 ± 0.76 | 83.23 ± 0.64 | 77.95 ± 0.68 | 36.30 ± 1.98 | 38.87 ± 3.66 |
| GT-sep | 87.32 ± 0.39 | 52.18 ± 0.80 | 92.29 ± 0.47 | 82.52 ± 0.92 | 78.05 ± 0.93 | 36.66 ± 1.63 | 40.31 ± 3.01 |
| CO-GNN | 89.44 ± 0.50 | **54.20 ± 0.34** | 97.35 ± 0.63 | 84.84 ± 0.96 | 75.97 ± 0.89 | 39.39 ± 2.76 | 41.14 ± 5.40 |
| O(d)-NSD | 80.41 ± 0.72 | 42.76 ± 0.54 | 92.15 ± 0.84 | 78.83 ± 0.76 | 69.69 ± 1.46 | 35.79 ± 3.34 | 37.93 ± 2.24 |
| BuNN | 91.75 ± 0.39 | 53.74 ± 0.51 | 98.99 ± 0.16 | 84.78 ± 0.80 | 78.75 ± 1.09 | - | - |
| CSNN | **92.63 ± 0.50** | 52.07 ± 1.00 | **99.07 ± 0.25** | **85.45 ± 0.53** | **79.31 ± 1.22** | **41.18 ± 2.23** | **43.09 ± 3.17** |

Table 2: Accuracy for node classification datasets on the fixed splits of Pei et al. (2020). CSNN achieves the best results in 2 out of 4 datasets.

| Model | Texas | Wisconsin | Film | Cornell |
| Edge Homophily | 0.11 | 0.21 | 0.22 | 0.30 |
|---|---|---|---|---|
| GGCN | 84.86 ± 4.55 | 86.86 ± 3.29 | 37.54 ± 1.56 | 85.68 ± 6.63 |
| H2GCN | 84.86 ± 7.23 | 87.65 ± 4.98 | 35.70 ± 1.00 | 82.70 ± 5.28 |
| GPRGNN | 78.38 ± 4.36 | 82.94 ± 4.21 | 34.63 ± 1.22 | 80.27 ± 8.11 |
| FAGCN | 82.43 ± 6.89 | 82.94 ± 7.95 | 34.87 ± 1.25 | 79.19 ± 9.79 |
| MixHop | 77.84 ± 7.73 | 75.88 ± 4.90 | 32.22 ± 2.34 | 73.51 ± 6.34 |
| GCNII | 77.57 ± 3.83 | 80.39 ± 3.40 | 37.44 ± 1.30 | 77.86 ± 3.79 |
| Geom-GCN | 66.76 ± 2.72 | 64.51 ± 3.66 | 31.59 ± 1.15 | 60.54 ± 3.67 |
| PairNorm | 60.27 ± 4.34 | 48.43 ± 6.14 | 27.40 ± 1.24 | 58.92 ± 3.15 |
| GraphSAGE | 82.43 ± 6.14 | 81.18 ± 5.56 | 34.23 ± 0.99 | 75.95 ± 5.01 |
| GCN | 55.14 ± 5.16 | 51.76 ± 3.06 | 27.32 ± 1.10 | 60.54 ± 5.30 |
| GAT | 52.16 ± 6.63 | 49.41 ± 4.09 | 27.44 ± 0.89 | 61.89 ± 5.05 |
| MLP | 80.81 ± 4.75 | 85.29 ± 3.31 | 36.53 ± 0.70 | 81.89 ± 6.40 |
| FSGNN | 87.57 ± 4.86 | 87.65 ± 3.51 | 35.62 ± 0.87 | **87.30 ± 4.53** |
| GloGNN | 84.32 ± 4.15 | 87.06 ± 3.53 | 37.35 ± 1.30 | 83.51 ± 4.26 |
| ACMGCN | **87.84 ± 4.40** | 88.43 ± 3.22 | 36.28 ± 1.09 | 85.14 ± 6.07 |
| CO-GNN | 77.57 ± 5.41 | 83.73 ± 4.03 | 36.26 ± 3.74 | 72.70 ± 5.47 |
| Diag-NSD | 85.67 ± 6.95 | 88.63 ± 2.75 | 37.79 ± 1.01 | 86.49 ± 7.35 |
| O(d)-NSD | 85.95 ± 5.51 | 89.41 ± 4.74 | 37.81 ± 1.15 | 84.86 ± 4.71 |
| Gen-NSD | 82.97 ± 5.13 | 89.21 ± 3.84 | 37.80 ± 1.22 | 85.68 ± 6.51 |
| CSNN | 87.30 ± 5.93 | **90.00 ± 2.83** | **38.03 ± 1.12** | 81.62 ± 4.32 |

For the remaining datasets, we compare against the classical GCN, GAT and SAGE; the models specifically tailored for heterophilic data GGCN (Yan et al., 2022), Geom-GCN (Pei et al., 2020), H2GCN (Zhu et al., 2020), GPRGNN (Chien et al., 2020), FAGCN (Bo et al., 2021), FSGNN (Maurya et al., 2022), GloGNN Li et al. (2022), ACMGCN (Luan et al., 2022), and MixHop (Abu-El-Haija et al., 2019); and the models GCNII (Chen et al., 2020) and PairNorm (Zhao and Akoglu, 2020) designed to alleviate oversmoothing. We use the 10 fixed splits proposed by Platonov et al. (2023) and Pei et al. (2020). We refer to Appendix C for further implementation details.

**Results.** Table 1 and Table 2 show that CSNN is the best-performing method in 9 out of 11 datasets. These results highlight our model's capacity to deal with heterophilic graphs of different sizes and heterophily levels. We note CSNN often outperforms both NSD and CO-GNN in Table 1. While we report the results in Table 2 for completeness, we note they exhibit high variance — in accordance to the findings of Platonov et al. (2023), which highlight that the small scale of these datasets may incur unstable and statistically insignificant results.

## 6.3 GRAPH CLASSIFICATION

To assess the effectiveness of CSNNs on long-range tasks, we evaluate it on the peptides dataset from the Long Range Graph Benchmark Dwivedi et al. (2022). It is a dataset containing 15k graphs and two different tasks: peptides-func is a graph classification task, while peptides-struct is a regression one. We report average precision (AP) for peptides-func and mean absolute error (MAE) for peptides-struct.

**Setup.** We follow the experimental setup of Tönshoff et al. (2024), and tune the network hyperparameters keeping the $\sim 500k$ parameter budget proposed by Dwivedi et al. (2022) for fair comparison. We run the model using four different seeds and report mean and standard deviation of the evaluation metrics. The baselines are taken from Tönshoff et al. (2024) and we also compare against the results reported for BuNN by Bamberger et al. (2025).

**Results.** Our model achieves the best performance in the peptides-struct dataset, and second-best in the peptides-func, as shown in Table 3.

Table 3: Performance comparison of models on the peptides datasets.

| Model | peptides-func ↑ | peptides-struct ↓ |
|---|---|---|
| GCN | 68.60 ± 0.50 | 24.60 ± 0.07 |
| GINE | 66.21 ± 0.67 | 24.73 ± 0.17 |
| GatedGCN | 67.65 ± 0.47 | 24.77 ± 0.09 |
| DReW | 71.50 ± 0.44 | 25.36 ± 0.15 |
| SAN | 64.39 ± 0.75 | 25.45 ± 0.12 |
| GPS | 65.34 ± 0.91 | 25.09 ± 0.14 |
| G-ViT | 69.42 ± 0.75 | 24.49 ± 0.16 |
| Exphormer | 65.27 ± 0.43 | 24.81 ± 0.07 |
| BuNN | **72.76 ± 0.65** | 24.63 ± 0.12 |
| CSNN | 71.58 ± 0.80 | **24.32 ± 0.04** |

These results further strengthen our claims on the capacity of CSNNs to mitigate over-squashing and perform better in scenarios where long-range and under-reaching are known issues.

# 7 CONCLUSION

This work proposed Cooperative Sheaf Neural Networks, a novel SNN architecture that incorporates directionality in order to increase its efficiency by learning sheaves with conformal maps, allowing nodes to choose the optimal behavior in terms of information propagation with respect to its neighbors. We provided theoretical insights on how CSNN can alleviate over-squashing due to its capacity to smoothly modulate node behavior in information diffusion. We also validated its effectiveness on node and graph classification experiments on heterophilic graphs and long-range tasks.

**Limitations and Future Works.** While CSNN is not computationally more taxing than other SNNs, it is worth pointing that developing strategies to scale sheaf-based networks is a major research challenge. While we have used conformal maps to reduce the parameter complexity of restriction maps, we leave open the possibility that there are further ways to improve the scalability of CSNN. We also believe that efficient message-passing implementations could represent a step towards large-scale SNNs.

Another promising direction for future works is extending SNNs to cope with high-order structures like cell- and simplicial complexes, possibly allowing for more expressive models and promoting long-range communication with fewer layers.

# 8 ETHICS AND REPRODUCIBILITY STATEMENTS

**Ethics Statement.** We do not foresee immediate negative societal or ethical impacts of our work.

**Reproducibility Statement.** Aiming to secure reproducibility of our work, we provide proofs of our theoretical results and in experiment detail in Appendix A and Appendix C. Moreover, we will provide a public code once the review process is complete.

# ACKNOWLEDGEMENTS

We acknowledge the support by the Fundação Carlos Chagas Filho de Amparo à Pesquisa do Estado do Rio de Janeiro (FAPERJ) (SEI-260003/020348/2025, SEI-260003/020694/2025) and the Conselho Nacional de Desenvolvimento Científico e Tecnológico (CNPq) (404336/2023-0, 305692/2025-9, 408974/2025-7, 312068/2025-5). We also thank The Adjoint School 2022 that introduced cellular sheaves to Ana Luiza Tenório, which contributed towards making this work possible.

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

## A PROOFS

### A.1 PROOF OF PROPOSITION 3.1

*Proof.* Suppose $L_{\mathcal{F}}(\mathbf{X})_i$ does not depend of $\mathbf{x}_j$ for any $j$ neighbor of $i$. Since $L_{\mathcal{F}}(\mathbf{X})_i = \sum_{i,j \trianglelefteq e} \mathcal{F}_{i \trianglelefteq e}^{\top} (\mathcal{F}_{i \trianglelefteq e}\mathbf{x}_i - \mathcal{F}_{j \trianglelefteq e}\mathbf{x}_j)$, this means $\mathcal{F}_{i \trianglelefteq e}^{\top}\mathcal{F}_{j \trianglelefteq e}\mathbf{x}_j = 0$. Therefore, $\mathcal{F}_{j \trianglelefteq e}\mathbf{x}_j \in \ker(\mathcal{F}_{i \trianglelefteq e}^{\top})$, for any $j$ neighbor of $i$. Thus, $\mathcal{F}_{i \trianglelefteq e}\mathbf{x}_i = 0$ or $\mathcal{F}_{j \trianglelefteq e}\mathbf{x}_j = 0$, for every $j$.

Note that, $\mathcal{F}_{i \trianglelefteq e}\mathbf{x}_i = 0$ implies that $L_{\mathcal{F}}(\mathbf{X})_j = \sum_{j,i \trianglelefteq e} \mathcal{F}_{j \trianglelefteq e}^{\top}\mathcal{F}_{j \trianglelefteq e}\mathbf{x}_j$.

If $\mathcal{F}_{j \trianglelefteq e}\mathbf{x}_j = 0$ for every $j$, then $L_{\mathcal{F}}(\mathbf{X})_j = 0$. $\qquad\square$

### A.2 REMARK REGARDING PROPOSITION 3.1 AND NONLINEAR SHEAF LAPLACIAN

If the sheaf Laplacian is nonlinear as in Zaghen et al. (2024), i.e, $L_{\mathcal{F}}(\mathbf{X})_i = \sum_{i,j \trianglelefteq e} \mathcal{F}_{i \trianglelefteq e}^{\top}\phi_e (\mathcal{F}_{i \trianglelefteq e}\mathbf{x}_i - \mathcal{F}_{j \trianglelefteq e}\mathbf{x}_j)$, where $\phi_e : \mathcal{F}(e) \to \mathcal{F}(e)$ is a continuous function for each edge $e$, then saying that $L_{\mathcal{F}}(\mathbf{X})_i$ does not depend of $\mathbf{x}_j$ means $\mathcal{F}_{i \trianglelefteq e}^{\top}\phi_e\mathcal{F}_{j \trianglelefteq e}\mathbf{x}_j = 0$. Then the proof holds similarly to the above.

### A.3 PROOF OF PROPOSITION 4.1

*Proof.* We have that $(L_{\mathcal{F}}^{\text{in}})^{\top} L^{\text{out}}$ valued at a given vertex $i$ is:

$$\sum_{j \in N(i)} \left( \mathbf{T}_i^{\top}\mathbf{T}_i \left( \sum_{j \in N(i)} \left( \mathbf{S}_i^{\top}\mathbf{S}_i\mathbf{x}_i - \mathbf{T}_i^{\top}\mathbf{S}_j\mathbf{x}_j \right) \right) - \mathbf{T}_i^{\top}\mathbf{S}_j \left( \sum_{u \in N(j)} \left( \mathbf{S}_j^{\top}\mathbf{S}_j\mathbf{x}_j - \mathbf{T}_j^{\top}\mathbf{S}_u\mathbf{x}_u \right) \right) \right) \quad (9)$$

So $\mathbf{T}_i = 0$ (i.e. $i$ does not listen) implies $((L_{\mathcal{F}}^{\text{in}})^{\top} L_{\mathcal{F}}^{\text{out}}(\mathbf{X}))_i = 0$. If $i$ listens, but a certain neighbor $k$ does not broadcast, i.e., $\mathbf{T}_k = 0$, then $((L_{\mathcal{F}}^{\text{in}})^{\top} L_{\mathcal{F}}^{\text{out}}(\mathbf{X}))_i$ is

$$\sum_{j \in N(i) \setminus k} \left( \mathbf{T}_i^{\top} \mathbf{T}_i \left( \sum_{j \in N(i) \setminus k} (\mathbf{S}_i^{\top} \mathbf{S}_i \mathbf{x}_i - \mathbf{T}_i^{\top} \mathbf{S}_j \mathbf{x}_j) \right) - \mathbf{T}_i^{\top} \mathbf{S}_j \left( \sum_{u \in N(j) \setminus k} (\mathbf{S}_j^{\top} \mathbf{S}_j \mathbf{x}_j - \mathbf{T}_j^{\top} \mathbf{S}_u \mathbf{x}_u) \right) \right)$$

Since the sum does not go through the index $k$, $\mathbf{x}_k$ is not a component in $((L_{\mathcal{F}}^{\text{in}})^{\top} L_{\mathcal{F}}^{\text{out}}(\mathbf{X}))_i$. $\square$

## A.4 Proof of Proposition 4.2

*Proof.* Let $t = 1$, and fix a node $i$. Then we are essentially just using the composition described in Equation (9) (up to normalization and learnable weights). In the equation we have a sum running over all neighbors $j$ of $i$ and another sum running over all neighbors $u$ of each $j$. So $u$ can be a 2-hop neighbor of $i$ and we have that $i$ was updated with information from up to 2-hop neighbors. Similarly, the node $u$ is updated by up to 2-hop neighbors. Therefore, in the second layer $t = 2$, $i$ was updated with information from up to 4-hop neighbors.

If $t = n$, assume by induction that each node $i$ receives information from it $2n$-hop neighbors. In the next layer $n + 1$, $i$ will be updated by its $n$-update of its 2-hop neighbors. Let $j$ be a node in the 2-hop neighborhood of $i$. By the inductive hypothesis, $j$ receives information from its $2n$-hop neighbors, whose distance to $i$ is up to $2n + 2 = 2(n + 1)$, concluding the proof by induction. $\square$

## A.5 Proof of Proposition 4.3

*Proof.* Choose a path from $i$ to $j$. So there are $t - 1$ vertices between $i$ an $j$, say $v_1, ..., v_{t-1}$. In the first layer, let $\mathbf{S}_j$ and $\mathbf{T}_{v_{t-1}}$ be different of zero and all other source and target maps equal zero.

This results in $((\Delta_{\mathcal{F}}^{in})^{\top} \Delta_{\mathcal{F}}^{out})_k = 0$ for every $k \neq v_{t-1}$ and $((\Delta_{\mathcal{F}}^{in})^{\top} \Delta_{\mathcal{F}}^{out})_{v_{t-1}}$ depends only of $x_j$. So, except for $x_{v_{t-1}}$, the values $x_k$ are not updated.

In the second layer, let $\mathbf{S}_{v_{t-1}}$ and $\mathbf{T}_{v_{t-2}}$ different of zero, and all other maps equal zero. This results in $((\Delta_{\mathcal{F}}^{in})^{\top} \Delta_{\mathcal{F}}^{out})_k = 0$ for every $k \neq v_{t-1}$, where $((\Delta_{\mathcal{F}}^{in})^{\top} \Delta_{\mathcal{F}}^{out})_{v_{t-1}}$ depends only of the $x_{v_{t-1}}^{(1)}$ that was updated in the previous layer and depends only of $x_j$.

We continue this reasoning until the $t$-layer, in which we make $\mathbf{S}_{v_1}$ and $\mathbf{T}_i$ different of zero, and all other source maps equal zero. This results in $((\Delta_{\mathcal{F}}^{in})^{\top} \Delta_{\mathcal{F}}^{out})_k = 0$ for every $k \neq i$, where $((\Delta_{\mathcal{F}}^{in})^{\top} \Delta_{\mathcal{F}}^{out})_{v_1}$ depend only of the $x_{v_1}^{(t-1)}$, which going backwards depends only of the original $x_j$, up to transformations given by the target and source maps. $\square$

## B Additional Experiments

In this section, we provide other two experiments. The first is a real-world large dataset and the second is a noisy implementation of Example 4.4, to illustrate how Proposition 4.3 may work in practice.

**Real-world.** We run CSNN on the Penn94, a dataset with $41.554$ nodes, $1.362.229$ edges, and whose feature dimension is 5, against multiple baselines as reported in Li et al. (2022) to illustrate CSNN's performance in a larger *ambiguous* heterophilic (Luan et al., 2024) dataset. We highlight that CSNN also outperform baselines on Squirrel, another dataset classified as ambiguous heterophilic.

**Synthetic.** In the following, we consider a path graph with four nodes as in Example 4.4 where the features are initialized in a similar fashion of the TreeNeighborsMatch task, but we consider that all nodes have a number of "blue neighbors" to add noisy information along the path, while the source and target (nodes 3 and 0) have the same number of these neighbors. The goal is to transfer information from node 3 to node 0.

Figure 6 shows an actual run of CSNN in such graph: if the norm of $\mathbf{T}_i$ is close to zero, then the arrow pointing to $i$ is not draw. Analogously to $\mathbf{S}_i$. If both are close to zero, the node is isolated, but if none of them are we consider the following:

- If $\mathbf{T}_i \mathbf{S}_j$ (in the in-Laplacian) and $\mathbf{T}_i \mathbf{S}_j$ (in the out-Laplacian) are both close to 0, then the edge $(j, i)$ does not exist

- If $\mathbf{T}_i \mathbf{S}_j$ (in the out-Laplacian) and $\mathbf{T}_i^\top \mathbf{T}_i$ are both close to 0, then edge $(j, i)$ does not exist.

Table 4: Results on the Penn94 dataset.

| Models | Penn94 |
|---|---|
| Edge Homophily | 0.47 |
| MLP | $73.61 \pm 0.40$ |
| GCN | $82.47 \pm 0.27$ |
| GAT | $81.53 \pm 0.55$ |
| MixHop | $83.47 \pm 0.71$ |
| GCNII | $82.92 \pm 0.59$ |
| H2GCN | $81.31 \pm 0.60$ |
| WRGAT | $74.32 \pm 0.53$ |
| GPR-GNN | $81.38 \pm 0.16$ |
| ACM-GCN | $82.52 \pm 0.96$ |
| LINKX | $84.71 \pm 0.52$ |
| GloGNN | $85.57 \pm 0.35$ |
| CSNN | $\mathbf{86.00 \pm 0.39}$ |

We observe restriction maps going to zero in a different configuration than the one exhibited in Example 4.4 but still sending $x_3^{(0)}$ to node 0 in the second layer, while suppressing other node features.

Technically, the significance of a node $i$ over $j$ in a given layer $t$ is measured by the norm of the the term that multiplies $x_i^{(t)}$ in the expression of $((\Delta_{\mathcal{F}(t)}^{in})^\top \Delta_{\mathcal{F}(t)}^{out})_j$. For instance, in Figure 6, the restriction maps provided that 1 listens to 0 and no one else, so it can be affected only by $x_1^{(0)}$ and by $x_0^{(0)}$. A calculation shows that $x_1^{(0)}$ is multiplied by a quantity about 0.7 while $x_0^{(0)}$ is multiplied by 132, approximately. This guarantees that in layer 2, when the message goes from node 1 to 0, the feature $x_1^{(0)}$ is not relevant and, in practice, is ignored.

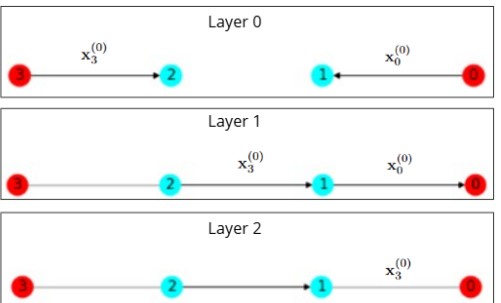

Figure 6: Illustration of running CSNN in a path graph with four nodes along three layers.

## C EXPERIMENT DETAILS

In this section, we provide the grid of hyperparameters used in the experiments. If the number of GNN layers is set to 0, we use an MLP with two layers to learn the restriction maps. Otherwise, we adopt a GraphSAGE architecture with the specified number of layers.

We also trained CO-GNN using the hyperparameter table from Finkelshtein et al. (2024), considering $\mu$ and $\Sigma$ as explicit hyperparameters instead of treating CO-GNN$(\mu, \mu)$ and CO-GNN$(\Sigma, \Sigma)$ as separate model variants.

All datasets except for roman-empire were treated as undirected graphs. For the roman-empire dataset, we found that using the stored list of edges was preferable to doubling the edges, since the graphs from Platonov et al. (2023) are stored as "directed" lists where elements as (0,2) and (2,0) are regarded as equivalent, for example.

All experiments were conducted on a cluster equipped with NVIDIA V100, A100, and H100 GPUs. The choice of GPU depended on the availability at the time of the experiments. Each machine was provisioned with at least 80 GB of RAM.

We also present some statistics of the heterophilic benchmarks in Table 7.

Table 5: Hyperparameter configurations used across heterophilic benchmarks.

| Parameter | roman-empire, amazon-ratings | minesweeper, tolokers, questions |
|---|---|---|
| sheaf dimension | $3, 4, 5$ | $3, 4, 5$ |
| # layers | 2–5 | 2–5 |
| # hidden channels | $32, 64$ | $32, 64$ |
| # of GNN layers | 0–5 | 0–5 |
| GNN dimension | $32, 64$ | $32, 64$ |
| dropout | 0.2 | 0.2 |
| input dropout | 0.2 | 0.2 |
| # epochs | 2000 | 2000 |
| activation | GELU | GELU |
| left weights | true, false | true, false |
| right weights | true, false | true, false |
| learning rate | 0.02 | 0.002, 0.02 |
| weight decay | $10^{-7}, 10^{-8}$ | $10^{-7}, 10^{-8}$ |

Table 6: Hyperparameter configuration used for NeighborsMatch.

| Parameter | NeighborsMatch |
|---|---|
| sheaf dimension | 2 |
| # layers | $r + 1$ |
| # hidden channels | 32 |
| # of GNN layers | $r + 1$ |
| GNN dimension | 32 |
| dropout | 0.0 |
| input dropout | 0.0 |
| activation | Id |
| left weights | true |
| right weights | true |
| layer norm | true |

Table 7: Statistics of the heterophilous datasets

| | roman-empire | amazon-ratings | minesweeper | tolokers | questions |
|---|---|---|---|---|---|
| nodes | 22662 | 24492 | 10000 | 11758 | 48921 |
| edges | 32927 | 93050 | 39402 | 519000 | 153540 |
| avg degree | 2.91 | 7.60 | 7.88 | 88.28 | 6.28 |
| node features | 300 | 300 | 7 | 10 | 301 |
| classes | 18 | 5 | 2 | 2 | 2 |
| edge homophily | 0.05 | 0.38 | 0.68 | 0.59 | 0.84 |
| adjusted homophily | -0.05 | 0.14 | 0.01 | 0.09 | 0.02 |
| metric | acc | acc | roc auc | roc auc | roc auc |

## D    COMPLEXITY AND RUNTIME OF CSNN

Using $d$ for dimension of the stalks, $h$ as the number of channels and $c = dh$, the complexity of our model is as follows:

- $O(d^2 |V|)$ for the embedding of graph features into the sheaf stalks;
- $O(|V| d^2 h) = O(|V| cd)$ when applying $W_1$;
- $O(|V| dh^2) = O(|V| ch)$ when applying $W_2$;

- $O(2|E|d^2h) = O(|E|cd)$ for the two sparse Laplacian-vector multiplication;
- $O(2d^3(|V| + |E|)) = O(d^3(|V| + |E|))$ for constructing the blocks of the Laplacians.

This gives a total of $O(|V|(c(d+h)+d^3)+|E|(cd+d^3))$. Since we use $1 \leqslant d \leqslant 5$, the stalk dimension contribution is small. We highlight that our code also contains a completely message-passing based implementation, that does not need constructing the Laplacian. This cheaper implementation yields a complexity of $O(c|V|(d + h) + |E|cd)$.

In the following we report the runtime of CSNN and the non-sheaf models on datasets of Table 1, as well as the improvement compared to the best baseline method.

Table 8: Runtime comparison on datasets from Platonov et al. (2023). We report the mean time in seconds per epoch, averaged over 10 epochs. CSNN has a similar runtime compared to these simple baselines, and presents a positive improvement on accuracy in general. The baselines achieving the best accuracy are highlighted **bold**.

| Model | roman-empire | amazon-ratings | minesweeper | tolokers | questions | squirrel | chameleon |
|---|---|---|---|---|---|---|---|
| GCN | 0.05s | 0.04s | 0.03s | 0.04s | 0.07s | **0.01s** | **0.02s** |
| SAGE | 0.07s | **0.04s** | 0.03s | 0.06s | 0.16s | 0.01s | 0.02s |
| GAT-sep | **0.09s** | 0.05s | **0.05s** | **0.12s** | 0.21s | 0.01s | 0.02s |
| GT-sep | 0.14s | 0.16s | 0.07s | 0.14s | **0.32s** | 0.02s | 0.01s |
| CSNN | 0.05s | 0.10s | 0.06s | 0.14s | 0.16s | 0.05s | 0.03s |
| Improvement | ↑4.37% | ↓2.90% | ↑5.50% | ↑2.00% | ↑1.61% | ↑4.33% | ↑5.38% |

We can see that sometimes CSNN is quicker than SAGE, and sometimes it is equal to GCN in terms of runtime. This might look counter-intuitive, but CSNN achieves its best performance with fewer parameters. For instance, for the roman-empire dataset, GCN has 2,269,714 parameters, while CSNN has 339,900, i.e. GCN has about 668% more parameters.

