# OpenReview forum: "Cooperative Sheaf Neural Networks"
_ICLR.cc/2026/Conference — ICLR 2026 Poster_

### Official Review · Reviewer_ZAJv · 2025-10-15

**Soundness:** 2
**Presentation:** 2
**Contribution:** 2
**Rating:** 4
**Confidence:** 3

**Summary:**

The paper argues that standard Sheaf Neural Networks (SNNs) on undirected graphs cannot realize node-level cooperative behavior (separating LISTEN vs PROPAGATE), because silencing incoming messages at a node also suppresses its outgoing influence.

**Strengths:**

### Clear motivation
* The limitation of undirected-sheaf diffusion for cooperative behavior is crisply identified and formalized. Proposition 3.1 explicitly shows PROPAGATE -> LISTEN coupling in classical SNNs; the directed-sheaf construction is a natural remedy

### Empirical breadth on heterophily
* Across 11 node-classification benchmarks (including the cleaned Squirrel/Chameleon), CSNN often outperforms both cooperative GNNs and prior SNNs

### Implementation details
* Parametrization for orthogonal maps, per-node conformal scaling, and a complexity discussion point to a careful engineering effort

**Weaknesses:**

### Novelty and Positioning
* Treating undirected edges as two directed edges plus per-node source/target maps is a natural extension but arguably an incremental one within the sheaf literature. The paper acknowledges related notions (e.g., quiver Laplacians), but the novelty boundary versus prior directed/sheaf constructions (and vector bundle variants on directed graphs) is not fully pinned down

### Theoretical Analysis
* Proposition 4.3 is an existence proof: there exist restriction maps for clean long-range, path-selective propagation. The paper does not analyze whether gradient-based training reliably finds such configurations in realistic, noisy data
* The paper notes their directed-sheaf Laplacians can have complex eigenvalues with negative real parts (unlike PSD Laplacians). Yet the stability of the discrete diffusionis not analyzed
* The model still performs local diffusion; the 2-hop per layer can accelerate reach but may also accelerate oversmoothing. There’s no theorem bounding oversmoothing for CSNN or showing improved expressivity beyond MPNN limitations

### Writing Clarifications
* A derivation/intuition for why this specific normalization (scalar multiples of identity under conformal maps) is preferred (vs. other block scalings) would help
* Regarding Proposition 4.3
  * The constructive path-only propagation is compelling but assumes the ability to zero out many maps across layers. In practice, with shared parameters and noisy optimization, how often does training approximate this regime?

**Questions:**

Please see the above weaknesses

---

> ### Author Response · Authors · 2025-11-20
> **Part 1/2**
>
> Thank you for reviewing our manuscript. We hope our answers below elevate your appraisal of our work. We did our best to address all your questions and concerns. Nonetheless, if you would like further clarifications, we will gladly engage further.
>
> > CSNN versus prior directed/sheaf constructions
>
> We appreciate this opportunity to highlight that there is no previous work with sheaves on directed graphs in the context of neural networks. The quiver Laplacian we mention in our manuscript was constructed for a different framework — feature selection on datasets, with no learning component involved.
>
> Moreover, although we could have generalized the sheaf Laplacian differently (e.g., replacing $T_i^{\top}S_j$ in Equation 7 by $S_i^{\top}T_j$ or used the quiver Laplacian), these choices do not support the cooperative behavior achieved by CSNN. This reinforces that the theory we developed to make our model sound is not incremental. We will check how to reformulate the Related works section to sharpen this point.
>
> > Regarding Proposition 4.3: “The paper does not analyze whether gradient-based training reliably finds such configurations in realistic, noisy data. [...] how often does training approximate this regime?”
>
> We appreciate your interest. The proof of Proposition 4.3 and Example 4.4 illustrate an ideal scenario where the message from $i$ can go to a distant node $j$ while ignoring all other nodes completely. In practice, the usual configuration is that the original feature of node $i$
> influence node $j$ more significantly than the original features of the other nodes, which lead to the same outcome. For instance, when we [run CSNN in a path graph with four node](https://anonymous.4open.science/r/Cooperative-783D/pathGraph.png), some restriction maps go to zero in a different configuration than the one exhibited in Example 4.4 but still sending $x_3^{(0)}$ to node 0 in the second layer, while suppressing other node features.
>
> Technically, the significance of a node $i$ over $j$ in a given layer $t$ is measured by the norm of the the term that multiplies $x_i^{(t)}$ in the expression of $((\Delta_{F(t)}^{in})^{\top}\Delta_{F(t)}^{out})_j$  (see Equation 9 to have a better grasp of the formalism). For instance, In the experiment for the path graph, the restriction maps provided that 1 listens to 0 and no one else, so it can be affected only by $ x_1^{(0)}$ and by $x_0^{(0)}$. A calculation shows that $x_1^{(0)}$ is multiplied by a quantity about $0.7$ while $x_0^{(0)}$ is multiplied by $132$, approximately. This guarantees that in layer 2, when the message goes from node 1 to 0, the feature  $x_1^{(0)}$ is not relevant and, in practice, is ignored.
>
> We will add an appendix to clarify this other configurations that allow message passage between distant nodes without relevant influence of intermediary nodes.
>
>
> > “the stability of the discrete diffusion is not analyzed”
>
> Indeed, since our Laplacians are not PSD matrices, complex eigenvalues with negative real parts may appear — making it difficult to characterize the limiting behaviour of repeatedly applying it to the node features. Nonetheless, it is also relevant to point out that the model does not apply the same Laplacian in an iterated manner. This is because a new sheaf is learned in each layer, resulting in a change to the Laplacian at every step. In other words, whether or not the diffusion process is stable has no direct impact on CSNN’s implementation.
>
> This motivated us to focus on consequences of the proposed model (Propositions 4.1, 4.2 and 4.3) instead of properties of the proposed directed Laplacians. Nonetheless, we agree that the stability analysis for non-PSD or Hermitian Laplacians is a significant (and challenging) area of research which deserves further investigation.

---

> ### Author Response · Authors · 2025-11-20
> **Part 2/2**
>
> > “the 2-hop per layer can accelerate reach but may also accelerate oversmoothing”
>
> While using a composition of in- and out- Laplacians allows for CSNNs to access 2-hop information, there are choices of restriction maps that make CSNNs selectively tend to 1-hop neighbors. For instance, in Figure 3, setting $T_2, S_3, S_1 \neq 0$ and $T_1=T_3=0$ ensures that all nodes will be updated with information from immediate (i.e., 1-hop) neighbors. In this case, we would not expect CSNNs to accelerate oversmoothing since it would present a behaviour closer to NSD, which is already known for handling oversmoothing on heterophilic data.
>
> Nevertheless, we exemplify that CSNN handles oversmoothing through the lens of the Dirichlet energy. As studied in [1,2] oversmoothing is associated with the decrease of the Dirichlet energy throughout the layers. For CSNN, the log of the Dirichlet energies in the last epoch of the train are:
> |model/layer | 4 | 8 | 16 | 32 |
> |-------------|-------------|-------------|-------------|-------------|
> | roman-empire | 13.31 | 13.61 | 22.69 | 30.67 |
> | minesweeper | 12.44 | 13.27 | 29.68  | 16.56 |
>
> This ensures that CSNN does not lose the ability to mitigate oversmoothing.
>
> > “A derivation/intuition for why this specific normalization (scalar multiples of identity under conformal maps) is preferred (vs. other block scalings) would help”
>
> Thanks for your comment. This normalization stems directly from our choice of using conformal maps. One compelling reason to use conformal maps is that computing their inverses becomes trivial — and so becomes normalizing the Laplacian. In this case, block scaling simplifies to a block matrix of scalars time identity — as we mention in the lines following Eq. 8. In addition, they also incur a smaller memory footprint than general restriction maps would. We will clarify that in the revised manuscript.
>
> [1] Bodnar, Cristian, et al. "Neural sheaf diffusion: A topological perspective on heterophily and oversmoothing in gnns." Neurips (2022)
>
> [2] Cai, Chen, and Yusu Wang. "A note on over-smoothing for graph neural networks." arXiv (2020)

---

> ### Comment · Reviewer_ZAJv · 2025-11-22
> **Thanks for your careful rebuttal**
>
> Dear Authors,
>
> Thank you for taking the time to address all of the raised concerns carefully. I would like to raise my score to 6.

---

### Official Review · Reviewer_o6mM · 2025-10-28

**Soundness:** 3
**Presentation:** 3
**Contribution:** 2
**Rating:** 6
**Confidence:** 3

**Summary:**

This paper extends Sheaf Neural Networks, a class of GNNs that use cellular sheaves to generalize message passing, to support cooperative communication between nodes. Classical SNNs effectively address heterophily and oversmoothing, but they lack the flexibility for nodes to independently choose how to exchange information (i.e., whether to propagate, listen, or isolate).
To overcome this, the authors propose the Cooperative Sheaf Neural Network (CSNN), which introduces directed cellular sheaves with separate in- and out-degree Laplacians, enabling asymmetric and selective communication.

**Strengths:**

Clear theoretical guarantees linked to long-range neighbors and over-squashing.

Empirical performance is compelling

**Weaknesses:**

Scalability of sheaf-based models on large-scale graphs remains to be tested.

**Questions:**

1. This paper seems to be an A+B work, with a combination of sheaf GNN + cooperative GNN, and some extension on directed graphs. And it is not well motivated or does not have some special designs for heterophily and over-squashing.

2. Why does the sheaf neural network have to achieve cooperative behavior?

3. "our model has the ability to reach longer distances." It is no surprise when you multiple your in-degree and out-degree sheaf Laplacian together. However, it is found that long-range information is harmful in many heterophilic datasets [1].

4. Missing comparison with some baseline models, e.g. ACMGCN [2], FSGNN [3], GloGNN [4]. More tests on malignant and ambiguous heterophilic datasets listed in [5]. Experiments on large scale datasets used in [4].




[1] Less is More: on the Over-Globalizing Problem in Graph Transformers. In Forty-first International Conference on Machine Learning 2024.

[2] Revisiting heterophily for graph neural networks. Advances in neural information processing systems. 2022 Dec 6;35:1362-75.

[3] Simplifying approach to node classification in graph neural networks. Journal of Computational Science, 62, 101695.

[4] Finding global homophily in graph neural networks when meeting heterophily. In International conference on machine learning (pp. 13242-13256). PMLR.

[5] The heterophilic graph learning handbook: Benchmarks, models, theoretical analysis, applications and challenges. arXiv preprint arXiv:2407.09618. 2024 Jul 12.

---

> ### Author Response · Authors · 2025-11-20
> **Part 1/2**
>
> We are thankful for your careful review and the opportunity to clarify our work. We respond to each of the raised concerns below, and we will be happy to engage in further discussion during the rebuttal period.
>
> > “...combination of sheaf GNN + cooperative GNN…”
>
> We appreciate the opportunity to clarify this. We highlight key points that make our model more sophisticated than combining the CO-GNN model and a sheaf neural network model.
>
> **There is no concept of environment or action networks:** CoGNN uses  environment and action networks to make the cooperation possible. Our CSNN model does not require such concepts.
>
> **Novel sheaf Laplacian:** There were no prior works on the Laplacian of cellular sheaves over directed graphs, we had to formulate one to propose our model. Importantly, there are many manners of generalizing the sheaf Laplacian, and not all choices support the cooperative behavior achieved by CSNN (i.e., not all forms of generalizing the sheaf Laplacian will provide that Proposition 4.1 holds). This reinforces that the theory we developed to make our model sound is not trivial.
>
> **CSNN does not use Gumbel Softmax:** Co-GNN uses a Gumbel Softmax estimator to learn categorical node’s actions, while CSNN learns actions through the restriction maps, approaching or distancing the null map.
>
> To illustrate the difference between CSNN and Co-GNN+SNN,  we have similarly ran experiments with Co-GNN + NSD and compared them both against NSD and our CSNN, retrieving the optimal NSD structure from BuNN’s paper. In this case, melding CoGNNs and NSD yields improvements (Co-NSD > NSD), but results are still significantly worse than CSNN’s.
> | Model  |  amazon-ratings | roman-empire | Minesweeper |
> |-----------------------|------------------|------------------|------------------|
> |NSD | 42.76 ± 0.54 | 80.41 ± 0.72 | 92.15 ± 0.84
> | Co-NSD | 51.46 ± 0.35 | 89.34 ± 0.29 | 94.28 ± 0.87 |
> | **CSNN** | **52.07 ± 1.00** | **92.63 ± 0.50** | **99.07 ± 0.25** |
>
> > Why does the SNN have to achieve cooperative behavior?
>
> Thank you for asking. SNNs do not have to achieve cooperative behaviour, but the cooperative framework shows improvements and substantially mitigate oversquashing, given the results in the TreeNeighborsMatch task. In this light, not being able to perform cooperation was a previous gap in the literature.
> Concretely, on the TreeNeighborsMatch task, BuNN reports 100% accuracy until r =6. Then it drops to 71% and 42% for r=7 and r=8, respectively, as reported in [1]. For NSD with orthogonal maps, we got 100% accuracy when r = 2,  91% for r = 3, then drops to 5% when r = 4. Our CSNN model shows 100% accuracy until r=8.
>
> As recommended by reviewer rWHL, we will add this comparison to NSD and BuNN in the synthetic experiment to better illustrate that the cooperative behavior present in CSNN improves learning on a controlled scenario of oversquashing.
>
>
> > “long-range information is harmful in many heterophilic datasets”
>
> We observe that CSNN can still access 1-hop information, e.g. in Figure 3, consider $T_2, S_3, S_1 \neq 0$ and $T_1=T_3=0$. Considering a single layer, Equation 9 (in Appendix A.2) provides that the node 2 will be updated with information from nodes 1 and 3 (in the 1-hop) but not from the node 4, which is in the 2-hop. Note that if $T_2, S_3, S_1 \neq 0$, while $S_4 = 0$ , we have another way of obtaining that the node 2 is influenced only by nodes in the 1-hop. Therefore, the model does not lose the possibility of only attending 1-hop information. We would gladly add the above discussion to the paper to clarify such matters.

---

> ### Author Response · Authors · 2025-11-20
> **Part 2/2**
>
> > Comparison to ACMGCN, FSGNN and GloGNN.
>
> Thank you for your suggestion. We compare CSNN with these 3 models, where the results for FSGNN and GloGNN were taken from [1] and we run ACMGCN using our grid.
>
>  Models | roman-empire | amazon-ratings | minesweeper | tolokers | questions | squirrel | chameleon |
> |-------------|-------------|-------------|-------------|-------------|-------------|-------------|-------------|
> FSGNN | 79.92 ± 0.56 | **52.74 ± 0.83** | 90.08 ± 0.70 | 82.76 ± 0.61 | 78.86 ± 0.92 | 35.92 ± 1.32 | 40.61 ± 2.97 |
> GloGNN | 59.63 ± 0.69 | 36.89 ± 0.14 | 51.08 ± 1.23 | 73.39 ± 1.17 | 65.74 ± 1.19 | 35.11 ± 1.24 | 25.90 ± 3.58 |
> ACMGCN | 64.34 ± 1.31 | 36.81 ± 3.59 | 82.54 ± 0.40 | 79.21 ± 0.47 |  53.16 ± 2.49 | 33.41 ± 2.15 | 36.24 ± 4.00 |
> **CSNN** | **92.63±0.50** | 52.07 ± 1.00 | **99.07 ± 0.25** | **85.45 ± 0.53** | **79.31 ± 1.22** | **41.18 ± 2.23** | **43.09 ± 3.17** |
>
>
> The results above show that CSNN consistently outperforms the new baselines. The only exception is the amazon-rating dataset. In the following, we used the datasets from Table 2. In this case, CSNN has the best performance in 2 of 4 datasets. It is worth mentioning that the variance on them is high, a phenomenon that was described in [1].  We appreciate your suggestions and will add these results to our revised manuscript.
>
>  Models | Texas | Wisconsin | Cornell | Film |
> |-------------|-------------|-------------|-------------|-------------|
> FSGNN | 87.57 ± 4.86 | 87.65 ± 3.51 | **87.30 ± 4.53** | 35.62 ± 0.87 |
> GloGNN | 84.32 ± 4.15 | 87.06 ± 3.53 | 83.51 ± 4.26 | 37.35 ± 1.30 |
> ACMGCN | **87.84 ± 4.40** | 88.43 ± 3.22 | 85.14 ± 6.07 | 36.28 ± 1.09 |
> **CSNN** | 87.30 ± 5.93 | **90.00 ± 2.83** | 81.62 ± 4.32 | **38.03 ± 1.12** |
>
>
> > More tests on malignant and ambiguous heterophilic datasets and large scale datasets
>
> In the above, Cornell, Wisconsin, Texas, Film, and roman-empire are malignant heterophilic datasets. We observe that we used squirrel-filtered, which is an ambiguous dataset. Now, we add Penn94 as a larger and ambiguous heterophilic dataset. We used the results reported in [2].
>
>  Models | Penn94
> |-------------|-------------|
> MLP | 73.61 ± 0.40 |
> GCN | 82.47 ± 0.27 |
> GAT | 81.53 ± 0.55 |
> MixHop | 83.47 ± 0.71 |
> GCNII | 82.92 ± 0.59 |
> H2GCN | 81.31 ± 0.60 |
> WRGAT | 74.32 ± 0.53 |
> GPR-GNN | 81.38 ± 0.16 |
> ACM-GCN | 82.52 ± 0.96 |
> LINKX | 84.71 ± 0.52 |
> GloGNN | 85.57 ± 0.35 |
> *CSNN* | **86.00 ± 0.39** |
>
> Overall, our results demonstrate CSNN outperforms the baselines on both malignant and ambiguous heterophilic data of different sizes. We will revise the manuscript to include this discussion, which we hope elevates your appraisal of our work.
>
> [1] Platonov et al. "A critical look at the evaluation of GNNs under heterophily: Are we really making progress?." ICLR
>
> [2] Finding global homophily in graph neural networks when meeting heterophily. In International conference on machine learning (pp. 13242-13256). PMLR.

---

### Official Review · Reviewer_rWHL · 2025-10-31

**Soundness:** 2
**Presentation:** 3
**Contribution:** 2
**Rating:** 4
**Confidence:** 4

**Summary:**

The work proposes a new Sheaf Neural Network model, called Cooperative Sheaf Neural Network (CSNN), with the goal of combining the benefits of Sheaf Neural Networks in tackling oversmoothing in GNNs and handling heterophilic data, with the property of performing selective communication between nodes, which is expected to lead to cooperative behaviour and avoid oversquashing problems. The model and its benefits are validated through synthetic and real-world experiments.

**Strengths:**

- The motivation, goal, and methodology are well formulated.
- The paper reads well, it has a nice structure, and the necessary background knowledge is well presented.

**Weaknesses:**

There are strong claims regarding (1) consistently outperforming SNNs and cooperative GNNs (both in introduction and in the experiments, see Results in Section 6.2), and (2) about not soccumbing to oversquashing (abstract). Looking at the results, it doesn't appear to improve substantially with respect to the competitors, so I would advise to reconsider the strength of the claims, which may lead to high expectations in the experiments.

Additionally, the introduction of a sheaf structure generally adds computational and runtime overhead compared to a GNN model, so its inclusion should be well justified in two respects: (1) whether the additional cost is outweighed by the gain in prediction accuracy, and (2) whether there is a real practical need for a sheaf-based model. Regarding (1), a discussion ideally comparing the model with non-sheaf baselines, is missing in the main part of the paper. Regarding (2), I would expect a convincing discussion of the advantages over other SNN methods in preventing oversquashing (due to the additional cooperative component), and over cooperative GNNs in mitigating oversmoothing and handling heterophilic datasets (thanks to the additional sheaf structure). These advantages are not evident from the experiments, as the relevant comparisons are either missing or the results/discussions are not sufficiently convincing.

**Questions:**

**General concerns**
- In your introduction, you mention that selective communication is a desirable property for SNN in order to tackle oversquashing. Recently, Nonlinear Sheaf Neural Networks have shown a similar behavior, selectively exploiting information of neighbors in complex node interactions [1]. Do you think there may be a connection between your method and the employment of a nonlinear Laplacian?
- After definition 2.3: "...however, when they publicly discuss this topic, they may prefer to not manifest their true opinion. ". In my understanding, since the edge stalks may be different from the node stalks, the individuals don't necessarily need to discuss the topics of the private opinion spaces. The topics in the public agreement space may also be different.
- It is not clear how your result in Proposition 4.3 relates to your discussion on the oversquashing behavior relying on the definitions in [2] and [3], which you reference after Example 4.4. These works formalize oversquashing as a bound on the Jacobian, as you also state, and this bound is strongly influenced by the presence of $A_{i,j}^{(t)}$. In Example 4.4, although you show that at each layer one node is influenced by only a single other node, when computing the Jacobian the update matrices $T$ and $S$ will still accumulate in the product across multiple layers. So, there is still a product of $O(t)$ matrices in the bound for the Jacobian. Wouldn’t this have the same effect as including the term  $A_{i,j}^{(t)}$? Could you please elaborate on this point, perhaps by explicitly showing how your method improves this bound compared to a sheaf model that does not perform selective communication?

**Experiments**
- It would be useful to have in Table 1 and 2 an homophily measure for each dataset, to intuitively understand the setting.
- I would recommend to rephrase the claims regarding the results. For example, in the "Results" of section 6.2, you state "We note CSNN often outperforms both NSD and CO-GNN by a significant margin", although looking at the results of Table 1 and 2, and considering the +/- confidence bound, the improvements are not as significant as stated.
- As mentioned in the weaknesses, I believe the experiments would benefit from a direct comparison between the proposed method and its non-sheaf (CO-GNN) and non-cooperative (NSD/BuNN) counterparts - for example, by including these models in the synthetic experiments of Section 6.1 and adding CO-GNN in Table 2 to provide a more complete comparison and a clearer demonstration of the claims.

**Writing/Typos**
- In related works, in the paragraph related to Cooperative GNNs: "... that chooses the cooperation the action each node takes". There seems to be an error in the structure of this sentence.
- In Table 2, last column, the second-best result is not highlighted with grey color.



[1] Zaghen et al., Sheaf Diffusion Goes Nonlinear: Enhancing GNNs with Adaptive Sheaf Laplacians (2024)

[2] Di Giovanni et al., On over-squashing in message passing neural networks: The impact of width, depth, and topology. (2023)

[3] Topping et al., Understanding over-squashing and bottlenecks on graphs via curvature (2022)

---

> ### Author Response · Authors · 2025-11-20
> **Part 1/2**
>
> We appreciate the reviewer’s thoughtful feedback. In the following, we tried our best to mitigate your concerns and answer all your questions and we will be happy to engage in further discussion during the rebuttal period.
>
> > Strong claims
>
> Thank you for your suggestion. We will revise the manuscript to alleviate claims such as the one you mention in section 6.2. Yet, we clarify that the phrase “CSNN can handle long-range interactions without succumbing to oversquashing” refers to the result obtained in the synthetic TreeNeighborsMatch task. As far as we know, CSNN is the first model to achieve 100% accuracy in such a task designed to measure the success of GNN models to deal with oversquashing in a controlled environment.
>
> > CSNN against non-sheaf models (accuracy and computational efficiency)
>
> Thank you for the opportunity to explore this important matter deeply. In the following we report the runtime of CSNN and the non-sheaf models on datasets of Table 1, recalling the accuracy for CSNN and the best non-sheaf model. As you will see, sometimes CSNN is quicker than SAGE and even equal to GCN on runtime. This might look counter-intuitive, but there is an explanation: CSNN achieves its best performance with fewer parameters. For instance, for the roman-empire dataset, GCN has 2269714 parameters while CSNN has 339900, i.e., GCN has about 668% more parameters.
>
>
> **roman-empire:**
>
> GCN: 0.05s/epoch
>
> SAGE: 0.07s/epoch
>
> GAT-sep: 0.09s/epoch - 88,75%
>
> GT-sep: 0.14s/epoch
>
> CSNN: 0.05s/epoch - 92.63% - about 4% increase
>
> **amazon-ratings:**
>
> GCN: 0.04s/epoch
>
> SAGE: 0.04s/epoch - 53.63% - about 3% increase
>
> GAT-sep: 0.05s/epoch
>
> GT-sep: 0.16s/epoch
>
> CSNN: 0.10s/epoch - 52.07%
>
> **minesweeper:**
>
> GCN: 0.03s/epoch
>
> SAGE: 0.03s/epoch
>
> GAT-sep: 0.05s/epoch - 93.91%
>
> GT-sep: 0.07s/epoch
>
> CSNN: 0.06s/epoch - 99.07% - about 5% increase
>
> **tolokers:**
>
> GCN: 0.04s/epoch
>
> SAGE: 0.06s/epoch
>
> GAT-sep: 0.12s/epoch - 83.78%
>
> GT-sep: 0.14/epoch
>
> CSNN: 0.14s/epoch - 85.45% - about 2% increase
>
> **questions:**
>
> GCN: 0.07s/epoch
>
> SAGE: 0.16s/epoch
>
> GAT-sep: 0.21s/epoch
>
> GT-sep: 0.32s/epoch - 78.05%
>
> CSNN: 0.16s/epoch - 79.31% - about 2% increase
>
>
> **squirrel**
>
> GCN: 0.01s/epoch - 39.47%
>
> SAGE: 0.01s/epoch
>
> GAT-sep: 0.01s/epoch
>
> GT-sep: 0.02s/epoch
>
> CSNN: 0.05s/epoch  - 41.18% - about 4% increase
>
> **chameleon**
>
> GCN: 0.02s/epoch - 40.89
>
> SAGE: 0.02s/epoch
>
> GAT-sep: 0.02s/epoch
>
> GT-sep:  0.01s/epoch
>
> CSNN:  0.03s/epoch - 43.09% - about 5% increase
>
>
> > CSNN against other SNN models on oversquashing task
>
> On the TreeNeighborsMatch task, BuNN reports 100% accuracy until r =6. Then it drops to 71% and 42% for r=7 and r=8, respectively, as reported in [1]. For NSD with orthogonal maps, we got 100% accuracy when r = 2,  91% for r = 3, then drops to 5% when r = 4. We appreciate your question and will report these values on the revised manuscript.
>
>
> > CSNN against cooperative +GNN on Table 2
>
>
> Thank you for this suggestion, we will add the following results on Table 2. Notably, CSNN achieves higher average accuracy for all datasets.
>
> |         Model         |        Film       |     Cornell     |      Texas      |   Wisconsin   |
> |-----------------------|------------------|------------------|------------------|-------------------|
> |      CO-GNN       | 36.26 ± 3.74 | 72.70 ± 5.47 | 77.57 ± 5.41 | 83.73 ± 4.03 |
> |      **CSNN**      | **38.03 ± 1.12** | **81.62 ± 4.32** | **87.30 ± 5.93** | **90.00 ± 2.83** |
>
> > Do you think there may be a connection between your method and the employment of a nonlinear Laplacian?
>
> Thanks for the relevant pointer. NLSD extends Bodnar et al.’s NSD by replacing the sheaf Laplacian $\delta^{\top} \delta$ by $\delta^{\top} \Phi \delta$, in which $\Phi$ is a non-linear transformation. That said, it is easy to extend Proposition 3.1 to NLSD and show that NLSD cannot _achieve cooperation_ either. The proof is similar to the one in the manuscript. We will include this extension in the revised manuscript.
>
> Yet, while not originally proposed for oversquashing, NLSD might also show gains (over NSD) in this area. Beyond that, we see as unlikely a formal connection between the two models since our Laplacians are linear while the one in NLSD is not.
>
>
> > “The topics in the public agreement space…”
>
> We agree with your understanding. The point we wanted to highlight was that even when topics in the public agreement space are the same, the opinion expressed does not necessarily agree with the individual opinions. Nevertheless, we will rephrase that part to clarify the possibility that different topics may be discussed.

---

> ### Author Response · Authors · 2025-11-20
> **Part 2/2**
>
> > How Proposition 4.3 relates to discussion on oversquashing
>
> Thank you for the opportunity to improve our presentation in this part.
> In [2] and [3], the upper bound helps to understand the impact of the graph topology  by $A_{i,j}^t$ and the impact of the model by the term $c^t$. Thus, for a fixed graph, the term $A_{i,j}^t$ does not change but $c^t$ does. We failed to convey that while sensibility between two nodes in these works is described by a derivative, we directly describe sensibility between $i$ and $j$ as $i$ depending on $j$ while possibly ignoring all others (Proposition 4.3). To illustrate the relevance of this distinction, consider the path graph in Example 4.4: The derivative of $x_1^{(3)}$ with respect to $x_4^{(0)}$ accumulates a product of matrices S and T as you observed; however, the derivative of  $x_1^{(3)}$ with respect to $x_k^{(0)}$, for $k= 2,3$ will be zero since the right S and T are zero. In other words, node $1$ is more sensible to node $4$ than to the other nodes. In simple GNNs like a GCN, the derivative of $x_1^{(3)}$ with respect to $x_k^{(0)}$, for $k= 2,3$ would not be zero, so $1$ could be equally sensitive (or more) to the nodes in comparison to node $4$. We will update the manuscript accordingly to include a proper discussion about it.
>
> Addressing your specific question, explicitly comparing the bound of the Jacobian for CSNN and another sheaf model for a fixed graph means comparing their term $c^t$, which is not clear how to properly calculate in sheaf neural networks, since it depends on the learned restriction maps. Yet, the results in the TreeNeighborsMatch experiment point out that CSNN likely has a higher bound than other SNNs, since it presents improved performance.
>
>
> > homophily measure
>
> Thank you for pointing this out. We will add this information in the manuscript. Their  edge homophily level are:
>
> roman-empire: 0.05
>
> amazon-ratings: 0.38
>
> minesweeper: 0.68
>
> tolokers:  0.59
>
> questions:  0.84
>
> squirrel: 0.20
>
> chameleon: 0.23
>
> Texas: 0.11
>
> Wisconsin: 0.21
>
> Film: 0.22
>
> Cornell: 0.30
>
>
> > Typos
>
> 1.Thank you, it should be simply “that chooses the best action for each node”. We will correct this.
>
> 2.Thank you for observing it. We will put it in gray, following the pattern.
>
>
> [1] Bamberger, Jacob, et al. "Bundle Neural Network for message diffusion on graphs." ICLR 2025.
>
> [2] Di Giovanni et al., On over-squashing in message passing neural networks: The impact of width, depth, and topology. (2023)
>
> [3] Topping et al., Understanding over-squashing and bottlenecks on graphs via curvature (2022)

---

### Official Review · Reviewer_4F1X · 2025-11-02

**Soundness:** 4
**Presentation:** 3
**Contribution:** 4
**Rating:** 8
**Confidence:** 4

**Summary:**

The authors propose an approach for learning cooperative behavour between nodes, by treating undirected edges between nodes as pairs of directed edges and introducing directed sheaf neural networks to effectively deal with them.

**Strengths:**

The work is well motivated, addressing a clear limitation of sheaf neural networks in modeling cooperation patterns between nodes.

The proposed solution has solid theretical grounds.

Synthetic results clearly confirm the ability to mitigate oversquashing, and an extensive evaluation on real-world datasets shows competitive performance wrt the state of the art.

**Weaknesses:**

Experimental results on the classical datasets for heterophilic analysis (Table 2) show very marginal improvements (considering the high variance, most likely none of these is significant). This is not a novelty, and questions the appropriateness of these datasets as benchmarks (as Platonov et al already pointed out). I encourage the authors to briefly discuss this aspect, so as to direct further research towards more appropriate evaluation benchmarks.

**Questions:**

Is it possible to have a high-level illustration clarifying the problem with plain SNN and the advantage of CSNN? This would help interested readers not familiar with the math behind SNN to gather the intuition behind the approach.

---

> ### Author Response · Authors · 2025-11-20
>
> Thank you for your thoughtful review and appraisal of our work. We did our best to address your question and improve our work. We will gladly engage further if you feel that any points require additional clarification.
>
> > “...classical datasets for heterophilic analysis (Table 2) show very marginal improvements ...”
>
> We appreciate the opportunity to reinforce this matter. As you noted, we anticipated higher variance for the Table 2 datasets based on Platanov et al.'s analysis but we included them as they remain standard in the community. We will add a discussion about this to the manuscript.
>
> > “...high-level illustration…”
>
>  Thank you for asking. In the [following illustration](https://anonymous.4open.science/r/Cooperative-783D/CSNNandNSD.png) we exhibit the effect of a certain node unable to listen, which breaks the communication completely in the Neural Sheaf Diffusion (NSD) model.  In contrast, our CSNN model still allows the same node to propagate. We will incorporate this into the manuscript. Please, let us know if this addresses your concern.

---

### Author Response · Authors · 2025-12-02
**Author Final Remarks**

Dear reviewers, (senior) ACs, and program chairs,

Thank you for your time, valuable feedback, and service to the community.

We have addressed all the reviewers' suggestions for additional experiments and discussion in our revised manuscript. A summary of these changes is provided below, and all modifications are highlighted in blue in the updated PDF.

- We incorporated another interpretation of the space of public agreement (line 123), as suggested by reviewer rWHL.
- We revised the manuscript to moderate strong claims and correct typos.
- We improved the explanation regarding our normalization in line 245.
- We added Figure 3 as a high-level illustration to clarify the limitations of NSD and highlight the advantages of our CSNN model.
- In lines 314-316, we clarified that, in addition to the 2-hop access proven by Proposition 4.2, our model also has the flexibility to access 1-hop neighbors per layer. This flexibility helps the model to achieve, in practice, the behaviour described in Proposition 4.3,  where distant nodes can communicate selectively, bypassing intermediate nodes. To sharpen this point, we have also provided a synthetic experiment in Appendix B.
- We complemented the connection between Proposition 4.3 and the bound of the Jacobian $|\partial x_i^{(t)}/\partial x_j^{(0)}|$  in lines 337-340.
- We further commented on the Quiver Laplacian (line 358).
- We also demonstrated that CSNN surpasses other sheaf models, specifically BuNN and NSD, on the classic synthetic task designed to measure oversquashing, NeighborsMatch (Section 6.1).
- In Section 6.2 we enhanced the presentation of CSNN performance of real-world datasets: we exhibit the edge homophily level of all datasets, added 3 new baselines (FSGNN, GloGNN, and ACMGCN), and make explicit the discussion from Platonov. et al about the high variances on Table 2.
- We append an argument to show that Proposition 3.1, which stated the non-cooperative behaviour of NSD, extends to its nonlinear version, i.e., the Nonlinear Sheaf Diffusion model cannot exhibit a cooperative behaviour in the sense we discuss in the paper (Appendix 4.2)
- We tested CSNN in Penn94, an ambiguous heterophilic dataset with 41.554 nodes and 1.362.229 edges. CSNN outperformed the baselines, showing its capacity to deal with larger heterophilic datasets.
- Finally, we presented Table 8 to compare the runtime of CSNN against GCN, SAGE, GAT-sep, and GT-sep.  For some datasets, CSNN is quicker than SAGE and even equal to GCN on runtime, which we justify with CSNN's ability to achieve its best performance with fewer parameters.

It is unfortunate that the leak problem has prevented further discussion and we could not hear back from reviewers ``4F1X``, ``rWHL``, ``o6mM``. However, we are glad that Reviewer ``ZAJv`` acknowledged that our rebuttal has “addressed all of the raised concerns carefully”, and we thank them for increasing their score from 4 to 6.

We believe acting on reviewers’ feedback has significantly strengthened this work, and we would like to reiterate our appraisal of their contributions.

Best,

The Authors

---

### Meta-Review · Area_Chair_QUZW · 2026-01-06

**Summary:**

**Summary:**
This paper introduces Cooperative Sheaf Neural Networks (CSNNs) which are a combination of Cooperative Graph Neural Networks and Sheaf Neural Networks, and are introduced to address limitations in both. The authors do this by equipping a CGNN with cellular sheaves over directed graphs with separate in and out degree laplacians. The authors validate this model through extensive theoretical analysis, synthetic experiments, and benchmark datasets.

**Rationale:**
The paper makes a clear and well-motivated contribution. This contribution is theoretically well characterized and the experiments are extensive, careful, and convincing. While the idea isn't particularly novel, it does move the state of the field forward. Further, the reviewer consensus is generally positive with one champion. Therefore, I'm recommending to accept.

**Reviewer Concerns:**

4f1x:
- Illustration: The reviewer wanted a diagram, authors provided
- Heterophilic variance: Authors agreed that some of these heterophilic datasets are problematic. This isn't a major problem given that there are so many datasets presented

rwhl:
- Multiple paper edits: authors moderated strong claims, added missing homophily measures, fixed typos
- Nonlinear sheaf connection: Authors added a new proposition
- NSD on synthetic datasets: The authors added this analysis and saw big wins
- Jacobian bound comparison: Authors point out that this depends on the learned restriction map which is hard to characterize generally. This was dropped


o6mm:
- Missing baselines: Authors expanded the baselines
- Larger datasets: Authors provided
- Concerns about long-range poisoning: The model can learn to only focus on 1 hop if necessary
- Runtime tradeoff: This doesn't scale to large graphs. Authors dropped this point.

zajv:
- Dirichlet energy: Authors show that CSNN isn't dissipative
- Normalization: Authors were able to justify where this comes from
- Quiver laplacians: Authors expanded on the difference between a quiver and directed laplacian
- Stability: Authors acknowledged that this was an open question
- Expressivity: Analysis didn't focus on the WL hierarchy

**Reviewer Scores:**

- 4f1x, 8 -> 8. Started as a strong supporter and had no concerns to bump up further
- rwhl, 4 -> 5. The major concerns were addressed. Their concern about Jacobian bounds was incomplete, but this concern feels fiddly in light of the synthetic results
- o6mm, 6 -> 6, Score was already high. The concerns that existed were not significant enough to, once addressed, reach an 8
- zajv, 4 -> 6, The reviewer explicitly raised their score already

---

### Decision · Program_Chairs · 2026-01-26

Accept (Poster)